# Nanog safeguards early embryogenesis against global activation of maternal β-catenin activity by interfering with TCF factors

**Mudan He**[1,2], **Ru Zhang**[1,2], **Shengbo Jiao**[1,2], **Fenghua Zhang**[1,2], **Ding Ye**[1,2], **Houpeng Wang**[1,2], **Yonghua Sun**[1,2]*

1 State Key Laboratory of Freshwater Ecology and Biotechnology, Institute of Hydrobiology, Innovation Academy for Seed Design, Chinese Academy of Sciences, Wuhan, China, 2 College of Advanced Agricultural Sciences, University of Chinese Academy of Sciences, Beijing, China

* yhsun@ihb.ac.cn

**Data Availability Statement:** All relevant data are within the paper and its Supporting Information files.

## Abstract

Maternal β-catenin activity is essential and critical for dorsal induction and its dorsal activation has been thoroughly studied. However, how the maternal β-catenin activity is suppressed in the nondorsal cells remains poorly understood. Nanog is known to play a central role for maintenance of the pluripotency and maternal -zygotic transition (MZT). Here, we reveal a novel role of Nanog as a strong repressor of maternal β-catenin signaling to safeguard the embryo against hyperactivation of maternal β-catenin activity and hyperdorsalization. In zebrafish, knockdown of *nanog* at different levels led to either posteriorization or dorsalization, mimicking zygotic or maternal activation of Wnt/β-catenin activities, and the maternal zygotic mutant of *nanog* (MZ*nanog*) showed strong activation of maternal β-catenin activity and hyperdorsalization. Although a constitutive activator-type Nanog (Vp16-Nanog, lacking the N terminal) perfectly rescued the MZT defects of MZ*nanog*, it did not rescue the phenotypes resulting from β-catenin signaling activation. Mechanistically, the N terminal of Nanog directly interacts with T-cell factor (TCF) and interferes with the binding of β-catenin to TCF, thereby attenuating the transcriptional activity of β-catenin. Therefore, our study establishes a novel role for Nanog in repressing maternal β-catenin activity and demonstrates a transcriptional switch between β-catenin/TCF and Nanog/TCF complexes, which safeguards the embryo from global activation of maternal β-catenin activity.

## Introduction

The Wnt/β-catenin signaling pathway, known as the canonical Wnt signaling pathway, is highly conserved during evolution. It plays crucial roles in embryonic development, organogenesis, tissue homeostasis, self-renewal and differentiation of stem cell, reproduction, and carcinogenesis [1–5]. Decades of studies have shown that the central scheme of the Wnt/β-catenin pathway is to stabilize the transcription coactivator β-catenin and protect it from phosphorylation-dependent degradation [6,7]. The Wnt/β-catenin pathway is well known for its "on/off" regulation model. In the presence of Wnt ligand, a receptor complex forms between

**Funding:** This work has received funding from the National Natural Science Foundation of China under grant No 31721005 and 31671501 to YS, and 31702323 to MH, from the National key R&D Program of China under grant No 2018YFA0801000 to YS, the Youth Innovation Promotion Association of Chinese Academy of Sciences to YS and the State Key Laboratory of Freshwater Ecology and Biotechnology under grant No 2019FBZ05 to YS. The funders had no role in study design, data collection and analysis, decision to publish, or preparation of the manuscript.

**Competing interests:** The authors have declared that no competing interests exist.

**Abbreviations:** AES, amino-terminal enhancer of split; bp, base pair; co-IP, co-immunoprecipitation; En-*nanog*, Nanog homeodomain fusion with Engrailed 2; ES, embryonic stem; GP domain, glycine/proline-rich domain; gRNA, guide RNA; GroBD, Groucho/TLE binding domain; HDAC, histone deacetylase; hpf, hours post fertilization; Hwa, Huluwa; ICM, inner cell mass; iPS, induced pluripotent stem; LD, low dose; MD, moderate dose; MO, morpholino; MZ*nanog*, maternal zygotic mutant of *nanog*; MZT, maternal zygotic transition; MZ*tle3a*, maternal zygotic mutant of *tle3a*; MZ*tle3b*, maternal zygotic mutant of *tle3b*; *nanog*_FL, full length of Nanog; *nanog*_ΔC, C-terminal truncated Nanog; RT-qPCR, reverse-transcription quantitative PCR; TALEN, transcription activator-like effector nuclease; TCF, T-cell factor; *tcf7*_ΔGroBD, Groucho-binding domain deleted Tcf7; *tcf7*_ΔHMG, high mobility group (LEF1-binding domain) deleted Tcf7; *tcf7*_ΔβBD, β-catenin-binding domain deleted Tcf7; TLE, transducin-like enhancer of split; vp16-*nanog*, Nanog homeodomain fusion with Vp16; WISH, whole-mount in situ hybridization; WT, wild type; YSL, yolk syncytial layer; ZGA, zygotic genome activation.

Frizzled and LRP5/6, and Frizzled is recruited by Dvl which leads to LRP5/6 phosphorylation and Axin recruitment, which in turn disrupts Axin-mediated phosphorylation/degradation of β-catenin, allowing β-catenin to accumulate in the nucleus where it serves as a coactivator for T-cell factor (TCF) to activate Wnt-responsive genes [8,9]. In the absence of Wnt ligand, cytoplasmic β-catenin forms a complex with Axin, APC, GSK3, and CK1 and is phosphorylated by CK1 and GSK3, recognized by the E3 ubiquitin ligase subunit β-Trcp, and processed through ubiquitination and proteasomal degradation. At this point, the nuclear TCFs are believed to physically interact with the co-repressors, the Groucho (Gro) and transducin-like enhancer of split (TLE) family members, and act as transcriptional repressors [10–13]. Thus, the activity of Wnt/β-catenin pathway is considered to be directly related to the level of nucleus β-catenin and its interaction with co-activators (such as activator-type TCFs) as well as co-repressors (such as repressor-type Gro/TLE) [8,14].

The dorsal accumulation of maternal β-catenin proteins plays a pivotal role in the dorsal axis formation in embryonic development, and the factors and mechanisms controlling maternal nuclear β-catenin accumulation and activation have been thoroughly studied. It is generally believed that the maternally inherited β-catenin proteins are translocated from the vegetal pole to the future dorsal side of embryos in a microtubule-dependent manner [15,16]. It was reported that the nuclear β-catenin is stabilized and activated by Wnt ligands, such as Wnt11 in *Xenopus* [17] and Wnt8a in zebrafish [18]. However, a recent *wnt8a* knockout study suggests that the maternally deposited *wnt8a* mRNA is not required for maternal β-catenin activation in zebrafish [19]. Several studies also indicate that Wnt receptors are not essentially required for maternal β-catenin signaling activation, because knockdown or knockout of Wnt receptor LRP5 did not lead to any early developmental abnormality in mice and zebrafish [20,21], and dorsal overexpression of dominant-negative LRP6 did not perturb the axis formation in *Xenopus* [22]. A recent study shows that the maternal clearance of Dvl activities did not lead to any early dorsal defects in zebrafish [23], further indicating that the maternal β-catenin activation is independent of Wnt ligand-receptor mediated process. More recently, it is demonstrated that a novel membrane protein Huluwa (Hwa) interacts with Axin to interfere in the formation of β-catenin destruction complex, thus promoting the activity of maternal β-catenin and dorsal axis formation [24]. In zebrafish, it is well established that the maternal and zygotic β-catenin signals play opposing roles in regulation of dorsal development [25–30], in which the maternally expressed *β-catenin 2* (*ctnnb2*) mainly activates the dorsal organizer genes, such as *dharma* (*boz*) and *chordin* (*chd*), and in turn the zygotically expressed *β-catenin 1* (*ctnnb1*) together with *ctnnb2* is required for the repression of dorsal organizer. Therefore, the spatial-temporal activation of β-catenin signaling is essential for the proper dorsoventral axis formation.

In principle, the maternal β-catenin activity should be strictly restricted to the dorsal-most cells and be globally repressed in the nondorsal embryonic cells, because ectopic activation of the maternal β-catenin activity would lead to disrupted dorsoventral axis [31]. In contrast to numerous studies focusing on the proper activation of maternal β-catenin signaling, the dorsal restriction of β-catenin signals and its repression in nondorsal cells are poorly understood. For instance, although *axin1* is maternally expressed in zebrafish, the *axin1* mutant, *materblind* (*mbl*) only exhibits a zygotic Wnt/β-catenin activation phenotype in which eyes and telencephalon are reduced or absent [32]. A Wnt/β-catenin pathway antagonist Chibby has been shown to physically interact with β-catenin to repress Wnt signaling [33,34], whereas loss of Chibby only results in a lung development defect in mice and basal body formation and ciliogenesis defects in *Drosophila*. Another type of β-catenin repressor, *ctnnbip1* (β-catenin interacting protein, previously named ICAT) could bind to the C-terminal region of β-catenin to disturb the interaction between β-catenin and TCF, and overexpression of a dominant-negative

*ctnnbip1* in ventral side can induce secondary axis in *Xenopus* [35], suggesting that ventral repression of β-catenin activity is crucial for proper dorsoventral axis formation. However, no mutant analysis of *ctnnbip1* has been described. The protein lysine demethylase Kdm2a/b is reported to mediate the demethylation of β-catenin and to promote its ubiquitination and degradation, but knockdown of *kdm2a/b* only leads to loss of posterior structures in *Xenopus* [36], and *kdm2* homozygous mutants show no developmental defects in *Drosophila* [37,38]. In zebrafish, Lzts2, Amotl2, and Tob1 are reported to inhibit β-catenin transcriptional activity by physically associating with β-catenin and preventing the formation of β-catenin/LEF1 complexes, but no mutant phenotypes have been reported to show how these factors affect the dorsal axis formation [39–41]. Therefore, what factor and which kind of mechanism repress the global β-catenin activation in nucleus need to be further studied with genetic null mutants.

Nanog is a core factor for maintenance of pluripotency and self-renewal of embryonic stem (ES) cells [42,43]. The *nanog*-deficiency ES cells lost pluripotency, and *nanog* mutant mice showed defects on ectodermal development and inner cell mass (ICM) proliferation [44]. Although Nanog is not on the list of the classical Yamanaka factors for induced pluripotent stem (iPS) cells, it has been demonstrated to be a "master switch" in the acquisition of cell pluripotency [45]. In zebrafish, previous studies have shown that *nanog* is a pivotal maternal factor to mediate endoderm formation through the Mxtx2-Nodal signaling in the extraembryonic yolk syncytial layer [46], and to initiate the zygotic genome activation (ZGA) together with Pou5f3 and SoxB1 during maternal zygotic transition (MZT) [47,48]. Recently, by generating maternal -zygotic mutants of *nanog* (MZ*nanog*), 2 studies further proved that zebrafish *nanog* is primarily required for extraembryonic development [49], and it is crucial for embryonic architecture formation and cell survival [50].

In the present study, we independently generate 2 MZ*nanog* alleles of zebrafish and demonstrate that maternal Nanog interacts with maternally deposited activator-type TCF in embryonic cell nuclei, thereby safeguarding the embryo against ectopic formation of β-catenin/TCF transcriptional activation complex, which may induce hyperdorsalization of the embryo. Our study thus uncovers a novel repressive regulation system of maternal β-catenin activity by revealing Nanog as a transcriptional switch between Nanog/TCF and β-catenin/TCF complexes.

## Results

### Maternally expressed TLEs do not likely repress maternal β-catenin activity in nondorsal cells of zebrafish early embryo

In zebrafish early embryonic development, maternal β-catenin proteins are generally believed to be translocated to the nuclei of dorsal blastomeres and subsequently activate the expression of a series of zygotic genes for dorsal commitment [51–55]. In a screening for maternal factors, we happened to observe that the nuclear β-catenin localized not only at the dorsal-most blastomeres with a high amount but also in some ventral and lateral cells with a low amount in early zebrafish embryos from 128-cell stage to high stage (S1 Fig), with the most prominence at 512-cell stage (Fig 1A); just like that in *Xenopus*, low levels of nuclear β-catenin also present throughout the embryo at blastulae stage [56,57]. We carefully examined the maternal expression of *wnt8* (*wnt8a ORF1* and *wnt8a ORF2*) and *ctnnb2* during oogenesis and in unfertilized eggs by in situ hybridization. Both *ctnnb2* and *wnt8a* were shown to be maternally expressed in developing oocytes and in unfertilized eggs (Fig 1B), suggesting that a certain amount of β-catenin might be existing in the nuclei of all the blastoderm cells, and its transcriptional activity should be repressed in the nondorsal cells. Furthermore, we showed that overexpression of *ctnnb2* or *hwa* in nondorsal cells could induce ectopic expression of maternal β-catenin

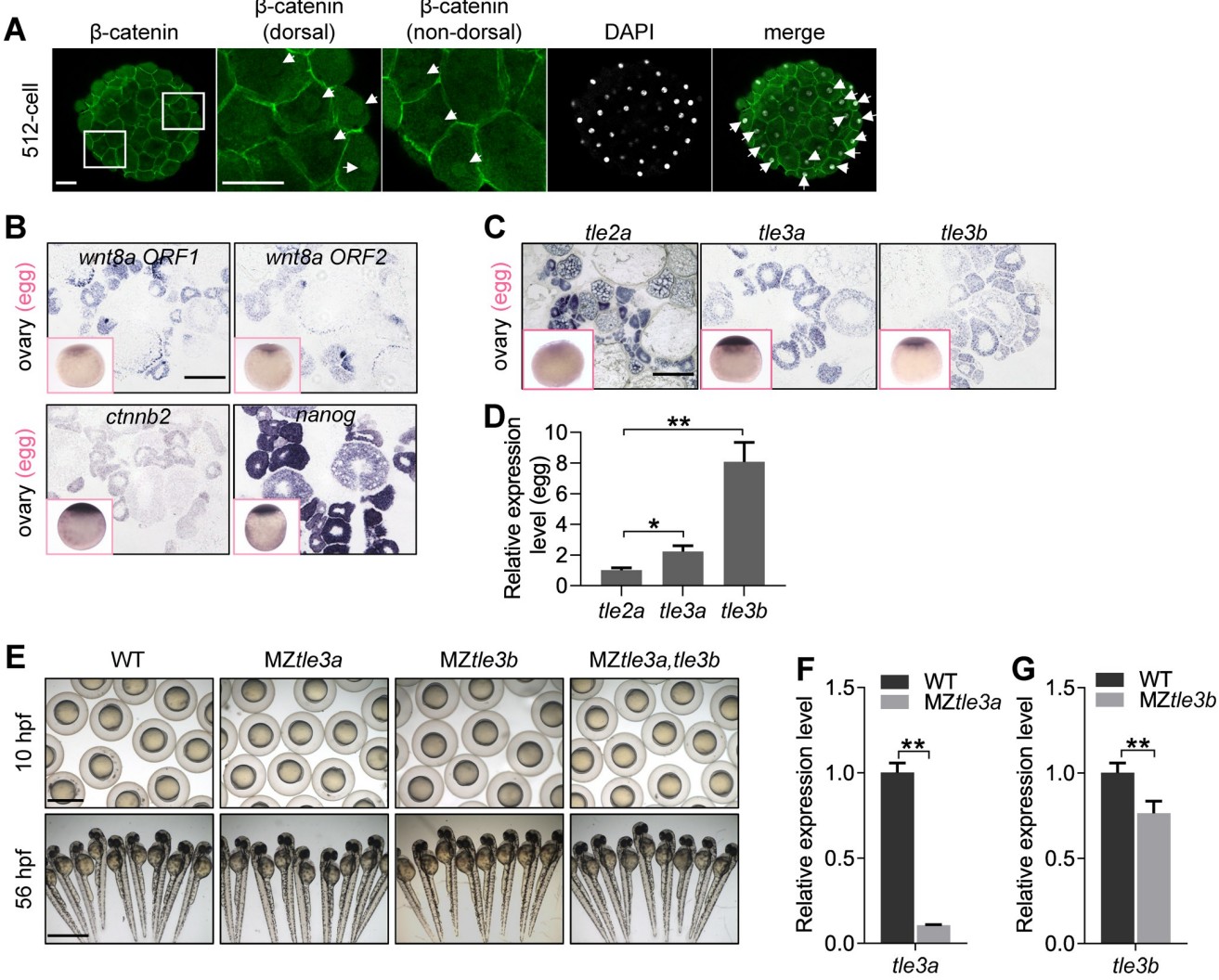

**Fig 1. Maternal TLEs do not likely contribute to the repression of maternal β-catenin activity.** (A) Detection of nuclear localization of maternal β-catenin in embryo at 512-cell stage by immunostainning against β-catenin. Signals were observed at animal view. Nuclei were co-stained with DAPI. Arrow heads indicate the nuclear accumulation of β-catenin. Scale bar, 50 μm. (B) In situ hybridization on cryosections of ovaries and WISH on unfertilized eggs (pink framed squares) showing *wnt8a1*, *wnt8a2*, *ctnnb2*, and *nanog* are maternally expressed during oogenesis and in unfertilized eggs. Scale bar, 100 μm. (C) In situ hybridization on cryosections of ovaries and WISH analysis of unfertilized eggs (pink framed squares) showing *tle2a*, *tle3a*, and *tle3b* are maternally expressed during oogenesis and in unfertilized eggs. Scale bar, 100 μm. (D) In comparison with *tle2a*, *tle3a*, and *tle3b* are significantly highly expressed in matured eggs as shown by RT-qPCR analysis. Error bars, mean ± SD, *$P < 0.05$, **$P < 0.01$. (E) The maternal-zygotic mutant of *tle3a* (MZ*tle3a*) or *tle3b* (MZ*tle3b*), or double mutant of *tle3a* and *tle3b* (MZ*tle3a*, *tle3b*), showed no early developmental defect. Scale bar, 1 mm. (F) RT-qPCR analysis showing mRNA expression level of *tle3a* was significantly reduced in MZ*tle3a* at 3 hpf. Error bars, mean ± SD, **$P < 0.01$. (G) RT-qPCR analysis showing mRNA expression level of *tle3b* was significantly reduced in MZ*tle3b* at 3 hpf. Error bars, mean ± SD, **$P < 0.01$. The *P* values in this figure were calculated by Student *t* test. The underlying data in this figure can be found in S1 Data. hpf, hours post fertilization; MZ*tle3a*, maternal-zygotic mutant of *tle3a*; MZ*tle3b*, maternal-zygotic mutant of *tle3b*; RT-qPCR, reverse-transcription quantitative PCR; TLE, transducin-like enhancer of split; WISH, whole-mount in situ hybridization; WT, wild type.

targets, *boz* and *chd* (S2A–S2C Fig), further supporting that the low level of ventrally located endogenous nuclear β-catenin activities should be repressed in early embryo.

Therefore, we were curious about how the ventrally distributed nuclear β-catenin activities are controlled. One possibility is the presence of certain antagonistic factors of β-catenin inside nucleus, e.g., Gro/TLE, which binds to suppressive TCF and interacts with histone deacetylases to maintain the chromatin in a transcriptionally inactive state [10–13,58]. Functional domain

aligning of all the TLE family genes in zebrafish (S3A Fig) showed that Tle2a, Tle3a (previous name, Groucho2), and Tle3b (previous name, Groucho1) are long Groucho/TLEs and potentially the antagonist of Wnt/β-catenin signaling [59], because they contain an N-terminal glutamine-rich Q domain and glycine/proline-rich (GP) domain which is involved in interactions with TCF and histone deacetylase (HDACs), a center DNA binding domain that medicates the binding of TLE with a variety of DNA sequences, and the C-terminal WD40 repeat domain that provides the primary binding site(s) for Engrailed via an eh1 motif. In contrast, Tle2b lacks the GP domain and center DNA binding domain, which seems unable to perform inhibition of Wnt/β-catenin signaling. Tle2c and Tle5 are the ortholog of human amino-terminal enhancer of split (AES), which only contains a Q-rich domain to compete with long Groucho/TLE homologs for Tcf binding, and serves as derepressors in some cases [59].

To figure out which long TLE might be the endogenous repressor of maternal Wnt/β-catenin signaling in zebrafish early embryo, we first analyzed the expression of *tle2a*, *tle3a*, and *tle3b* in ovary and eggs. Although we detected the maternal expression of *tle2a*, *tle3a*, and *tle3b* in developing oocytes (Fig 1C), *tle3a* and *tle3b* showed significant higher expression levels in matured eggs (Fig 1D). In recognition that Tle3a and Tle3b are the ortholog of human Grg2 that performs inhibition of Wnt/β-catenin signaling [60,61], we generated the maternal zygotic mutants of *tle3a* and *tle3b* by CRISPR/Cas9 technology (S3B and S3C Fig). To our surprise, neither the single or double maternal zygotic mutants of *tle3a* and *tle3b* showed any defects of early embryonic development (Fig 1E–1G). Then we knocked down the expression of *tle2a*, *tle3a*, and *tle3b* separately and in combinations. Phenotype observation showed knockdown of these *TLE* genes did not dorsalize the embryos (S3D–S3F Fig). All these data indicate that maternally expressed full-length *TLE*s, *tle2a*, *tle3a*, and *tle3b* do not likely contribute to the suppression of maternal β-catenin activity, and there might be other factors safeguarding the embryos against the activation of nuclear maternal β-catenin activity in nondorsal regions.

## Knockdown of *nanog* leads to dorsalization or posteriorization

During oogenesis, *nanog* is maternally expressed in the oocytes at different stages (Fig 1B). After fertilization, *nanog* mRNA is ubiquitously distributed in the blastoderm cells until 30% epiboly, dramatically decreased at shield stage and becomes undetectable from 75% epiboly (Fig 2A), implying an important role of zebrafish *nanog* during early development. To examine the function of *nanog* in early embryogenesis, we first utilized a previously published antisense morpholino (MO) targeting the translational start site of *nanog* mRNA to knock down *nanog* [46]. Intriguingly, injection of low-dose (LD) *nanog* MO (0.5 ng per embryo) mainly led to forebrain defects (Fig 2B), mimicking the zygotic overexpression of *wnt8a* [62]. We then increased the *nanog* MO dosage to 1.2 ng per embryo (moderate dose [MD]), and found that most of the embryos were strongly dorsalized (Fig 2B), resembling the observation in a recent study [63]. Both phenotypes (posteriorization and dorsalization) could be efficiently rescued by injection of the MO-binding site mismatched mRNA, indicating the specificity of phenotypes resulting from injection of *nanog* MO (S4A and S4B Fig). To clarify the different phenotypes after injection of different dosages of *nanog* MO, we performed a western blot to examine the Nanog protein levels at different stages in wild type (WT) and the embryos injected with LD or MD of MO (morphants). Our results showed that injection of LD *nanog* MO only led to partial reduction of Nanog protein expression level at 2 hours post fertilization (hpf) and complete elimination of Nanog protein at 4 hpf (Fig 2C and 2D), indicating that the endogenous Nanog proteins were eliminated after the time of ZGA. Whereas MD *nanog* MO injection led to complete absence of Nanog protein from 2 hpf (Fig 2C and 2D), indicating a block of the maternal activity of Nanog proteins.

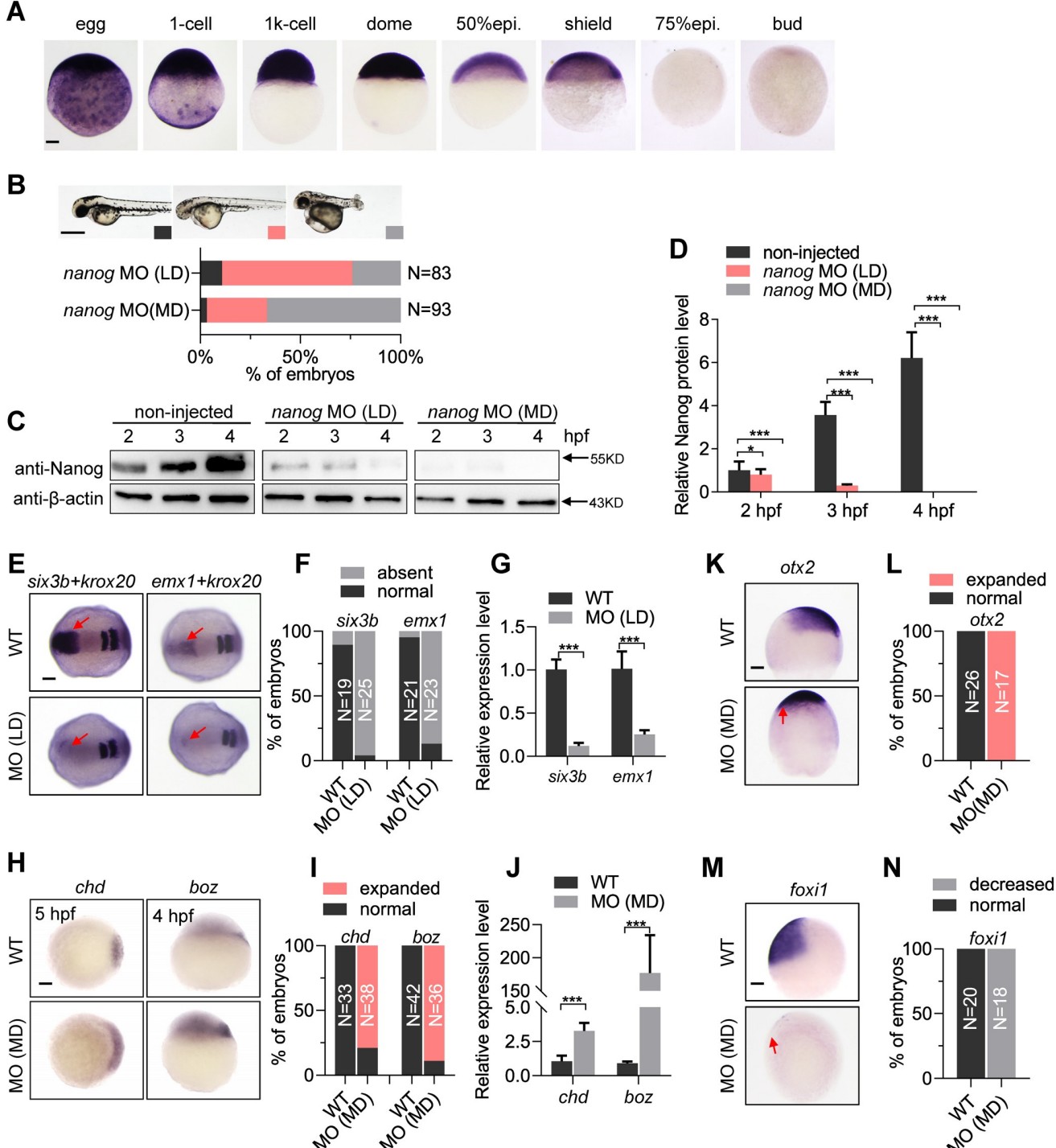

**Fig 2. Knockdown of *nanog* leads to dorsalization and posteriorization.** (A) WISH analysis showing *nanog* mRNA is maternally transcribed and vanishes at 75% epiboly stage. Scale bar, 100 μm. (B) Two different phenotypes are observed at 2 doses (0.5 ng/embryo, LD; 1.2 ng/embryo, MD) of *nanog* MO injected embryos, forebrain defect and dorsalization. Phenotype was observed at 36 hpf. N represents analyzed embryo number. Scale bar, 500 μm. (C) Western blot detection of Nanog in LD and MD *nanog* MO injected embryos. Nanog translation was blocked in all detected stages in the MD *nanog* MO injected embryos, and a low amount of Nanog protein can be detected at early stage in the LD *nanog* MO injected embryos. (D) Relative Nanog signal intensities in the western blot experiment (panel C). (E) WISH analysis showing expression of forebrain marker *six3b* and telencephalon marker *emx1* were absent in LD MO injected embryos. *krox20* was used as a stage-control marker. Red arrows indicate the expression region of *six3b* or *emx1*. Scale bar, 100 μm. (F) Statistical analysis of the embryos in panel E. N represents analyzed embryo number. (G) RT-qPCR analysis of *six3b* and *emx1* in *nanog* morphants and WT embryos. Error bars, mean ± SD, ***$P < 0.001$. (H) WISH analysis showing 2 maternal β-catenin targets, *boz* and *chd*, were up-

regulated in MD MO injected embryos. Scale bar, 100 μm. (I) Statistical analysis of the embryos in panel H. N represents analyzed embryo number. (J) RT-qPCR analysis of *chd* and *boz* in *nanog* morphants and WT embryos. Error bars, mean ± SD, ***P* < 0.001. (K) WISH analysis showing expression of dorsal neuroectoderm marker *otx2* was expanded in LD *nanog* MO injected embryos. Red arrow indicates the ventral expansion of *otx2* signals. Scale bar, 100 μm. (L) Statistical analysis of the embryos in panel K. N represents analyzed embryo number. (M) WISH analysis showing the expression of ventral epidermal ectoderm marker *foxi1* was eliminated in LD *nanog* MO injected embryos. Red arrow indicates the ventral absence of *foxi1* signals. Scale bar, 100 μm. (N) Statistical analysis of the embryos in panel M. N represents analyzed embryo number. *foxi1* and *otx2* were detected at 90% epiboly stage, *six3b* and *emx1* were detected at 2-somite stage, *chd* was detected at 5 hpf, and *boz* was detected at 4 hpf. The *P* values in this figure were calculated by Student *t* test. The underlying data in this figure can be found in S1 Data. hpf, hours post fertilization; LD, low dose; MD, moderate dose; MO, morpholino; RT-qPCR, reverse-transcription quantitative PCR; WISH, whole-mount in situ hybridization; WT, wild type.

Whole-mount in situ hybridization (WISH) analysis further confirmed the different phenotypes resulting from different dosages of MO injection. After injection of LD *nanog* MO, the expression of forebrain marker *six3b* and telencephalon marker *emx1* were nearly absent (Fig 2E–2G), mimicking the zygotic activation of Wnt/β-catenin signaling [18,64]. However, in the MD *nanog* morphants, the expression levels and expression territories of maternal β-catenin signaling targets *boz* and *chd* were strongly increased (Fig 2H and 2I), mimicking the dorsalization phenotype resulting from the ectopic activation of maternal β-catenin signaling (S2 Fig and ref [18]). The hyperactivation of maternal β-catenin targets, *chd* and *boz*, was confirmed by reverse-transcription quantitative PCR (RT-qPCR) analysis (Fig 2J). At midgastrula stage, the MD morphants showed strong expansion of dorsal ectoderm labeled by *otx2* (Fig 2K and 2L) and shrinkage of ventral ectoderm labeled by *foxi1* (Fig 2M and 2N), further demonstrating the dorsalization phenotype in the MD *nanog* morphants. Taken together, the above results suggest that elimination of Nanog activities before ZGA or after ZGA led to dorsalization or posteriorization, mimicking the maternal or zygotic activation of Wnt/β-catenin signaling, respectively.

## Knockdown of *nanog* activates maternal and zygotic Wnt/β-catenin signaling

To study the relationship between Nanog and Wnt/β-catenin signaling pathway, we performed a series of genetic interaction experiments. First, we compared the LD *nanog* morphants with the zygotic Wnt/β-catenin signaling elevated embryos. The LD *nanog* morphants showed forebrain defects, similar to the *wnt8a* (1 pg per embryo) overexpressed embryos or the *tcf7l1a* (previous name: *tcf3a* or *headless*) depleted embryos (*tcf7l1a* MO, 1.6 ng per embryo; Fig 3A) [18,64]. We then titrated the dosages of *nanog* MO (160 pg per embryo), *wnt8a* mRNA (0.1 pg per embryo), and *tcf7l1a* MO (800 pg per embryo) to obtain normal brain patterning in the injected embryos. However, when *nanog* MO was co-injected with *wnt8a* mRNA or *tcf7l1a* MO with the same dosages, a majority of embryos developed with the headless phenotype (Fig 3B), suggesting the genetic interaction between *nanog* and zygotic Wnt/β-catenin signaling. To further investigate the crosstalk between Nanog and Wnt/β-catenin signaling, we performed rescue experiments. In the LD *nanog* morphants, the expression of zygotic Wnt target gene *sp5l* was expanded to the animal pole and ventral ectoderm, and the expression of Wnt antagonist *dkk1b* and *frzb* was decreased, in comparison with WT (Fig 3C and 3D), whereas injection of *wnt8a* MO largely restored the expression of those genes in *nanog* morphants (Fig 3C and 3D). These expression alterations were further confirmed by RT-qPCR analysis (Fig 3E). More strikingly, the expression of 2 telencephalon markers, *six3b* and *emx1*, which were absent in the LD *nanog* morphants, were recovered after *wnt8a* knockdown (Fig 3F–3H). All these data indicate that Nanog negatively regulates Wnt/β-catenin signaling, and the LD *nanog* morphants could be partially rescued by knocking down *wnt8a*.

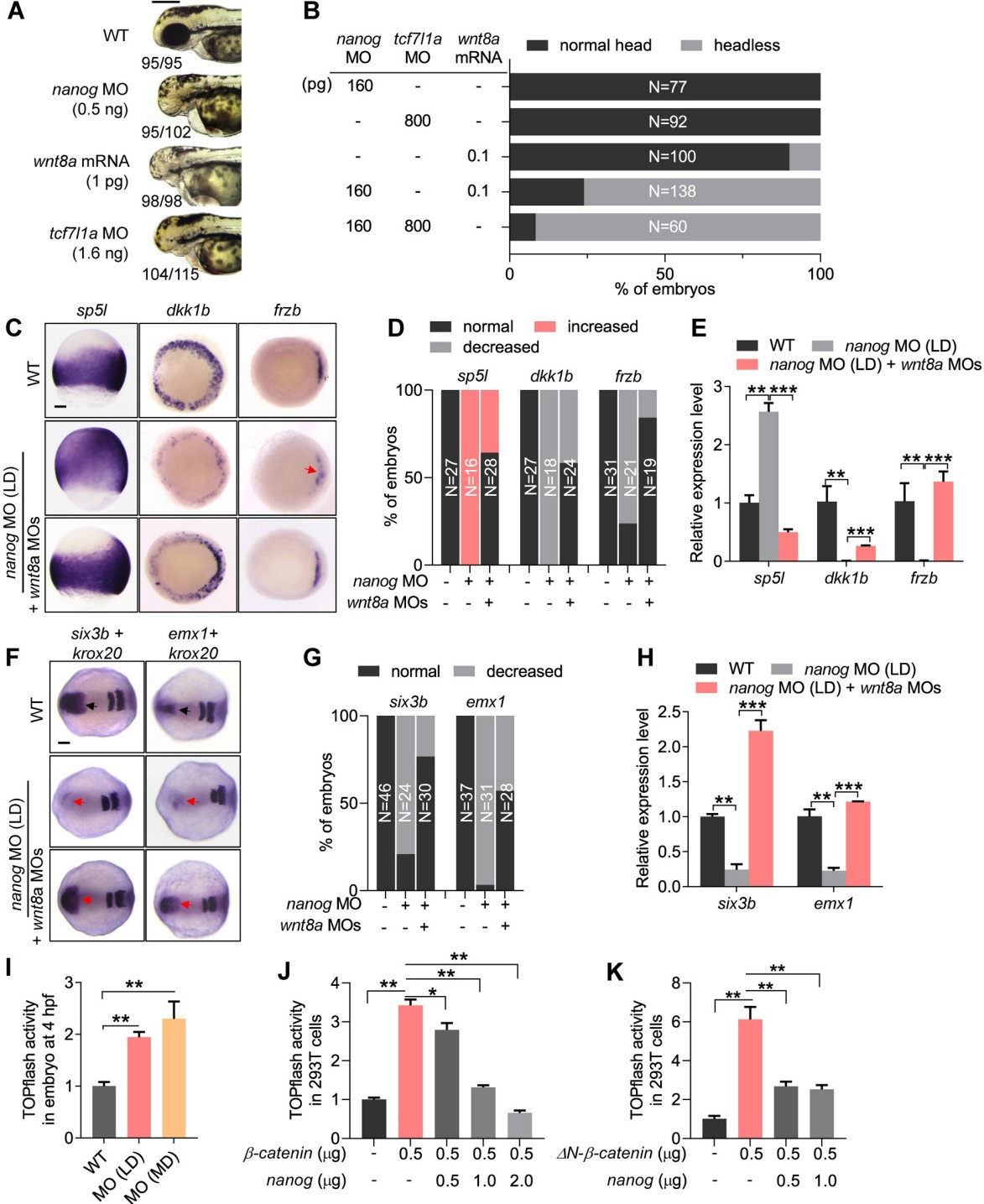

**Fig 3. _Nanog_ negatively regulates Wnt/β-catenin signaling.** (A) The embryos injected with LD _nanog_ MO (0.5 ng) exhibit the similar phenotypes—telencephalon defect—with _wnt8a_ mRNA (1 pg per embryo) overexpressed embryos or _tcf7l1a_ MO (1.6 ng per embryo) knocked down embryos. Phenotype was observed at 72 hpf. The numbers below the morphology pictures mean number of embryos showing representative phenotype/total number of embryos. Scale bar, 500 μm. (B) Embryos injected with titrated LDs of _nanog_ MO (160 pg), _wnt8a_ mRNA (0.1 pg), or _tcf7l1a_ MO (800 pg) showed no obvious defect, respectively, and co-injection of _nanog_ MO with _wnt8a_ mRNA or _tcf7l1a_ MO at the same doses resulted in forebrain truncation (headless). N represents analyzed embryo number. (C) WISH analysis showing the expression of zygotic Wnt target genes; _sp5l_ was up-regulated in _nanog_ morphant, whereas Wnt antagonist _dkk1b_ and _frzb_ were reduced, and this expression defect can be restored by knockdown of _wnt8a_. Scale bar, 100 μm. (D) Statistical analysis of the embryos in panel C. N represents analyzed embryo number. (E) Relative mRNA levels of _sp5l_, _dkk1b_, and _frzb_ in _nanog_ morphants and

rescued embryos examined by RT-qPCR. Error bars, mean ± SD, **$P < 0.01$, ***$P < 0.001$. (F) WISH analysis showing the forebrain defect in *nanog* morphant could be rescued by knockdown of *wnt8a1* and *wnt8a2*. Scale bar, 100 μm. (G) Statistical analysis of the embryos in panel F. N represents analyzed embryo number. (H) Relative mRNA level of *six3b* and *emx1* in *nanog* morphants and rescued embryos examined by RT-qPCR. Error bars, mean ± SD, **$P < 0.01$, ***$P < 0.001$. *sp5l* was detected at 75% epiboly, *frzb* and *dkk1b* were detected at 6 hpf, *six3b* and *emx1* were detected at 2-somite stage, *krox20* was used as stage control. (I) TOPflash analysis showing β-catenin transcriptional activity was up-regulated in both of LD and MD of *nanog* MO injected embryos at 4 hpf. Error bars, mean ± SD, **$P < 0.01$. (J) TOPflash assay showing co-transfection of Nanog inhibited the up-regulated β-catenin transcriptional activity induced by β-catenin in a dose-dependent manner in HEK293T cells. Error bars, mean ± SD, *$P < 0.05$, **$P < 0.01$. (K) TOPflash analysis showing co-transfection of Nanog inhibited the up-regulated β-catenin transcriptional activity induced by ΔN-β-catenin (a constitutively activated type of β-catenin) in HEK293T cells. Error bars, mean ± SD, **$P < 0.01$. The *P* values in this figure were calculated by Student *t* test. The underlying data in this figure can be found in S1 Data. HEK293T cells, human embryonic kidney 293T cells; hpf, hours post fertilization; LD, low dose; MD, moderate dose; MO, morpholino; RT-qPCR, reverse-transcription quantitative PCR; WISH, whole-mount in situ hybridization; WT, wild type.

We then investigated the repressive role of Nanog on Wnt/β-catenin signaling activity by in vivo and in vitro TOPflash assays [65]. In developing embryos at 4 hpf, a stage following ZGA, knockdown of *nanog* with LD or MD of *nanog* MO resulted in dose-dependent up-regulation of Wnt/β-catenin signaling activity (Fig 3I), inconsistent with the activation of maternal β-catenin signaling at this stage (Fig 2H–2J). In 293T cells, the TOPflash activity was also significantly increased after transfection of a β-catenin expression construct. However, when the cells were co-transfected with different amounts of Nanog, the TOPflash activity showed significant reduction in a dose-dependent manner (Fig 3J). We then performed a similar TOPflash assay in the cells transfected with ΔN-β-catenin, which can sustainably enter the nucleus to active the transcriptional activity [36]. Co-transfection of Nanog could still significantly repress the β-catenin transcriptional activity resulting from overexpression of ΔN-β-catenin in a dose-dependent manner (Fig 3K). All these results indicate that Nanog could effectively repress the transcriptional activity of nucleus-located β-catenin.

## Maternal β-catenin activity is hyperactivated in maternal zygotic mutants of *nanog*

In order to fully characterize the role of *nanog* in early embryonic development, we generate *nanog* mutants using transcription activator-like effector nuclease (TALEN)-mediated mutagenesis as described in our previous study [66]. We identified 2 alleles, 2–base pair (bp) deletion (−2) and 1-bp insertion (+1; S5A Fig). The 2-bp deletion resulted in the frame-shift of the open reading frame, and the 1-bp insertion results in premature termination at the target site and encodes a truncated Nanog of 18 amino acids (S5A Fig). The 2 alleles were named as *nanog*$^{ihb97/ihb97}$ (2-bp deletion) and *nanog*$^{ihb98/ihb98}$ (1-bp insertion), respectively. The zygotic mutant of *nanog* (Z*nanog*$^{ihb97}$ and Z*nanog*$^{ihb98}$) did not show any embryonic defect and grew up to adulthood normally (S5B Fig). We then continued to in-cross the zygotic mutant and obtained the maternal -zygotic mutant of *nanog* (MZ*nanog*). Compared with WT and Z*nanog* embryos, the MZ*nanog*$^{ihb97}$ embryos failed epiboly movement, and all cells stacked at the animal pole, and the MZ*nanog*$^{ihb98}$ embryos exhibited a less severe phenotype. Most of the embryos from both alleles of MZ*nanog* died within 24 hpf (S5B Fig). We obtained maternal mutant of *nanog* (M*nanog*$^{ihb97}$) by mating a MZ*nanog*$^{ihb97}$ female with WT male. M*nanog*$^{ihb97}$ embryos are phenotypically identical to MZ*nanog*$^{ihb97}$, implying that the early development was mainly regulated by maternally deposited *nanog* mRNA (S5B Fig).

We compared the transcript level of *nanog* in MZ*nanog*$^{ihb97}$ and MZ*nanog*$^{ihb98}$. In MZ*nanog*$^{ihb97}$ embryos, the maternal *nanog* mRNAs were absent, and the zygotic transcription of *nanog* appeared to happen at 1,000-cell stage but quickly vanished at 30% epiboly (S5C Fig). In MZ*nanog*$^{ihb98}$ embryos, the mutated *nanog* mRNA was present from 2-cell stage until 50%

epiboly but disappeared at 90% epiboly (S5C Fig). Because of the presence of a low level of mutated *nanog* transcripts and less penetrance of phenotype in MZ*nanog*[ihb98], we used MZ*nanog*[ihb97] (MZ*nanog*) for the subsequent experiments.

To verify that the Nanog protein was completely absent in MZ*nanog*, we detected the protein level of Nanog by western blot. In WT embryos, Nanog protein were detected from the 64-cell stage, reached a peak at sphere stage, and decreased rapidly during gastrulation. After 75% epiboly stage, Nanog protein was no longer detected, whereas the Nanog protein could not be detected in both types of MZ*nanog* embryos (Fig 4A). Immunostaining analysis of Nanog further confirmed that the Nanog is mainly localized in the cell nucleus, and there was no Nanog expression in MZ*nanog* embryos (Fig 4B). These data demonstrated that Nanog was completely depleted in MZ*nanog* mutant.

Just like the *nanog* morphants, the MZ*nanog* mutants showed strong activation of maternal β-catenin activity and hyperdorsalization, as indicated by expanded expression region of *boz* and *chd* (Fig 4C and 4D) and the significant decreased expression of *bmp2b*, *bmp7*, and *vent* (S6A and S6B Fig). When we injected a LD of *ctnnb2* (*β-catenin2*) mRNA (100 pg per embryo) into WT or MZ*nanog* embryos, the MZ*nanog* showed a dramatic expansion of *boz* and *chd*, although the LD of *ctnnb2* mRNA only slightly increased the expression of *boz* and *chd* in WT embryos (Fig 4C and 4D). The up-regulation of maternal β-catenin transcriptional targets, *boz* and *chd*, was further confirmed by RT-qPCR assay in MZ*nanog* embryos (Fig 4H). All these indicate that the maternal β-catenin activity is strongly activated in MZ*nanog* embryos, which leads to hyperdorsalization.

It has been shown that maternal and zygotic Wnt/β-catenin signals play opposing roles in regulating early dorsoventral patterning of zebrafish [25]. The dorsal organizing center formation depends on the maternal expression of *ctnnb2*, whereas both *ctnnb1* and *ctnnb2* are required for repressing the dorsal organizer genes, *boz* and *chd*, via the zygotic Wnt/β-catenin targets, *vox*, *vent*, and *ved* [26–30]. Therefore, we were curious whether the dorsalization of MZ*nanog* embryos was due to the activation of maternal β-catenin signaling. We tried to rescue the MZ*nanog* embryos through interfering with the expression of *ctnnb1* or *ctnnb2* by MO-mediated knockdown. Morphologically, the developmental defects of MZ*nanog* at the gastrula stage was largely rescued by knockdown of *ctnnb2* but not *ctnnb1* (Fig 4E). At molecular level, knockdown of *ctnnb2* reduced the ectopic expression of *boz* and *chd*, but *ctnnb1* knockdown did not (Fig 4F and 4G), which was further confirmed by RT-qPCR analysis (Fig 4H). As a result, the expression reduction of ventrally expressed *bmp2b*, *bmp7*, and *vent* could be restored by knockdown of maternally expressed *ctnnb2* but not zygotically expressed *ctnnb1* (S6A Fig). TOPflash assay also showed that the elevation of β-catenin signaling activity in MZ*nanog* embryos at 4 hpf could be significantly reduced by knockdown of *ctnnb2* but not *ctnnb1* (Fig 4I). Therefore, we conclude that loss of maternally provided Nanog activity leads to hyperactivation of maternal β-catenin activity and dorsalization, which is mediated by *ctnnb2*.

## Depletion of Nanog does not affect the nuclear translocation of β-catenin

To determine whether the hyperactivation of maternal β-catenin activity in MZ*nanog* was due to the up-regulation of maternally expressed *wnt8a* or *ctnnb2*, we first checked the expression levels of those transcripts. By WISH analysis, we found that the expression of *wnt8a* (*wnt8a ORF1* and *wnt8a ORF2*) and *ctnnb2* appeared comparable between WT and MZ*nanog* in unfertilized eggs and in the 16-cell stage embryos (Fig 5A). By RT-qPCR analysis, we found that *wnt8aORF1* even showed significantly decreased expression level in the ovary and the embryos at 1-cell, 2 hpf, and 4 hpf of MZ*nanog* when compared with WT (Fig 5B and 5C).

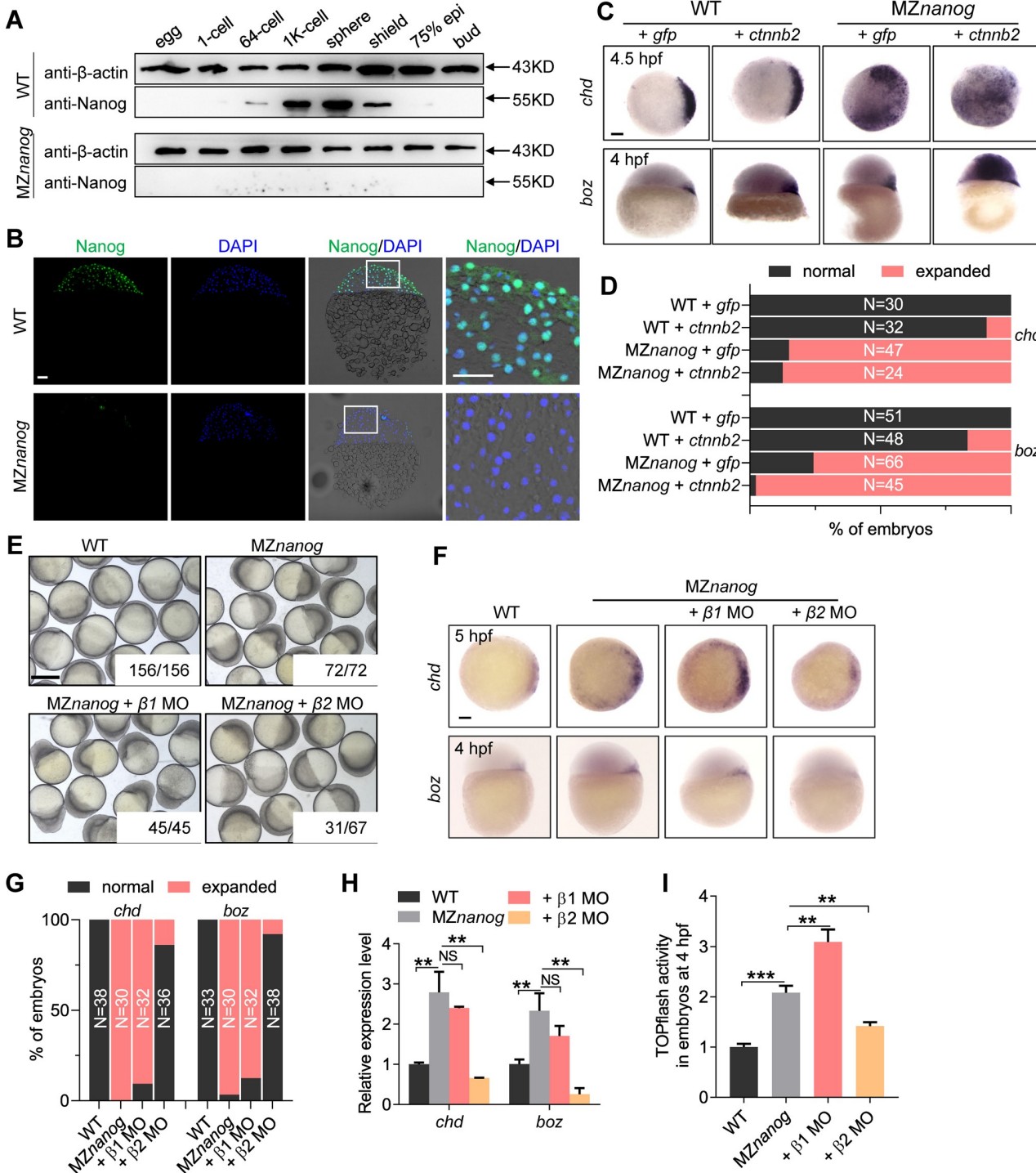

**Fig 4. Maternal β-catenin activity is hyperactivated in *nanog* mutant.** (A) western blot showing translation of Nanog protein totally disappeared in MZ*nanog*. Nanog protein can be detected as early as the 64-cell stage and vanished at 75% epiboly stage in WT embryos, whereas no Nanog protein was detected in MZ*nanog* mutant embryos. (B) Immunolocalization of Nanog on cryosections of WT and MZ*nanog* embryos at 4 hpf. Nanog is localized in the cell nuclei of WT embryos and disappeared in MZ*nanog* embryos. Nuclei were co-stained with DAPI. Scale bar, 50 μm. (C) WISH analysis showing the injection of low dose of *ctnnb2* mRNA (200 pg) induced slight up-regulation of *boz* and *chd* in WT embryos and induced massive expression of *boz* and *chd* in MZ*nanog* embryos. *boz* was detected at 4 hpf, and *chd* was detected at 4.5 hpf. Scale bar, 100 μm. (D) Statistical analysis of the embryos in panel C. N represents analyzed embryo number. (E) Knockdown of *ctnnb2* but not *ctnnb1* rescued the developmental defects of MZ*nanog* at gastrula stage. The numbers below the morphology pictures are the number of embryos showing representative phenotype/total number of embryos. Scale bar, 500 μm. (F) WISH analysis showing expression of *chd* and *boz* were expanded in MZ*nanog* embryos, and knockdown of *ctnnb2*

(*β2* MO) but not *ctnnb1* (*β1* MO) rescued these defects. *boz* was detected at 4 hpf, and *chd* was detected at 5 hpf. Scale bar, 100 μm. (G) Statistical analysis of the embryos in panel F. N represents analyzed embryo number. (H) Relative mRNA level of *boz* and *chd* in the embryos of WT, MZ*nanog*, MZ*nanog* co-injected with *ctnnb1* MO (+ *β1* MO) and MZ*nanog* co-injected with *ctnnb2* MO (+ *β2* MO). Error bars, mean ± SD, $^{**}P < 0.01$; NS means no significant difference. (I) TOPflash assay showing β-catenin transcriptional activity was significantly up-regulated in the embryos of MZ*nanog* compared with WT, and co-injection of *ctnnb2* MO (+ *β2* MO) but not *ctnnb1* MO (+ *β1* MO) into MZ*nanog* embryos significantly decreased the transcriptional activity. Error bars, mean ± SD, $^{**}P < 0.01$, $^{***}P < 0.001$. The *P* values in this figure were calculated by Student *t* test. The underlying data in this figure can be found in S1 Data. hpf, hours post fertilization; MO, morpholino; MZ*nanog*, maternal zygotic mutant of *nanog*; RT-qPCR, reverse-transcription quantitative PCR; WISH, whole-mount in situ hybridization; WT, wild type.

These data indicate that the hyperactivation of maternal β-catenin activity in MZ*nanog* is not due to the up-regulation of genes encoding Wnt ligands or β-catenin.

It is well known that the activation of canonical Wnt pathway leads to translocation of β-catenin into the cell nucleus [9,67]. We then aimed to test whether the hyperactivation of maternal β-catenin activity in MZ*nanog* was caused by increased accumulation of nuclear β-catenin. Firstly, we examined the level of nuclear β-catenin in MZ*nanog* at 4 hpf by western blot. Unlike the ratio of active β-catenin/total β-catenin remarkably increased in *wnt8a*-over-expressed embryos, the amount of nuclear β-catenin in MZ*nanog* embryos was even slightly decreased (Fig 5D and 5E). We further checked the nuclear β-catenin accumulation in MZ*nanog* embryos at the 512-cell stage by immunostaining and revealed that the nucleus-localized β-catenin in MZ*nanog* appeared comparable to that in WT (Fig 5F). Taken together, we demonstrate that the hyperactivation of maternal β-catenin activity in MZ*nanog* embryos is not due to the increased nuclear accumulation of β-catenin.

## N-terminal of Nanog is required for suppression of Wnt/β-catenin signaling

Because Nanog is a type of homeobox protein, we investigated whether Nanog functions as a transcriptional activator or repressor in zebrafish early development. We fused the homeodomain of Nanog with the transcriptional activator domain Vp16 or the transcriptional repressor domain of Engrailed 2 [68,69] to generate overexpression constructs of constitutive-activator type (Vp16-Nanog) or constitutive-repressor type (En-Nanog) of Nanog (S7A Fig). After injection of vp16-*nanog* mRNA into WT embryos, transcriptional targets of Nanog at ZGA, *mxtx2*, *blf*, and *mir-430*, showed increased expression, mimicking overexpression of WT *nanog* (S7B and S7C Fig). In contrast, injection of En-*nanog* mRNA led to decreased expression of those genes, mimicking *nanog* morphants (S7B and S7C Fig). On the other hand, the expression of *sod1*, a maternal mRNA targeted by miR-430 [70], was accumulated in *nanog* morphants and the En-*nanog* overexpressed embryos, indicating the defects of maternal mRNA clearance (S7B and S7C Fig). Further GFP-3xIPT-miR-430 reporter assay and RT-qPCR analysis of miR-430a and miR-430b confirmed the defects of MZT in MZ*nanog*, and the rescue effects by overexpression of full-length Nanog (Nanog_FL), vp16-*nanog*, and miR-430 (S7D–S7F Fig). All these demonstrate that Nanog serves as a transcriptional activator during MZT by activating the zygotic genes and miR-430, which can further clean the maternal mRNAs.

To understand the transcriptional activation effects of Nanog, we injected *nanog*_FL mRNA, vp16-*nanog* mRNA, and C-terminal truncated *nanog* (*nanog*_ΔC) mRNA into MZ*nanog* embryos to compare their rescue effects. Analysis of *mxtx2*, *blf*, *mir-430*, and *sod1* proved the defects of yolk syncytial layer (YSL) and defective MZT in MZ*nanog*, and overexpression of Nanog_FL (500 pg/embryo), Nanog_ΔC (25 pg/embryo), or Vp16-Nanog (250 pg/embryo) could successfully rescue the defects of YSL development and MZT in MZ*nanog* (Fig 6A–6C),

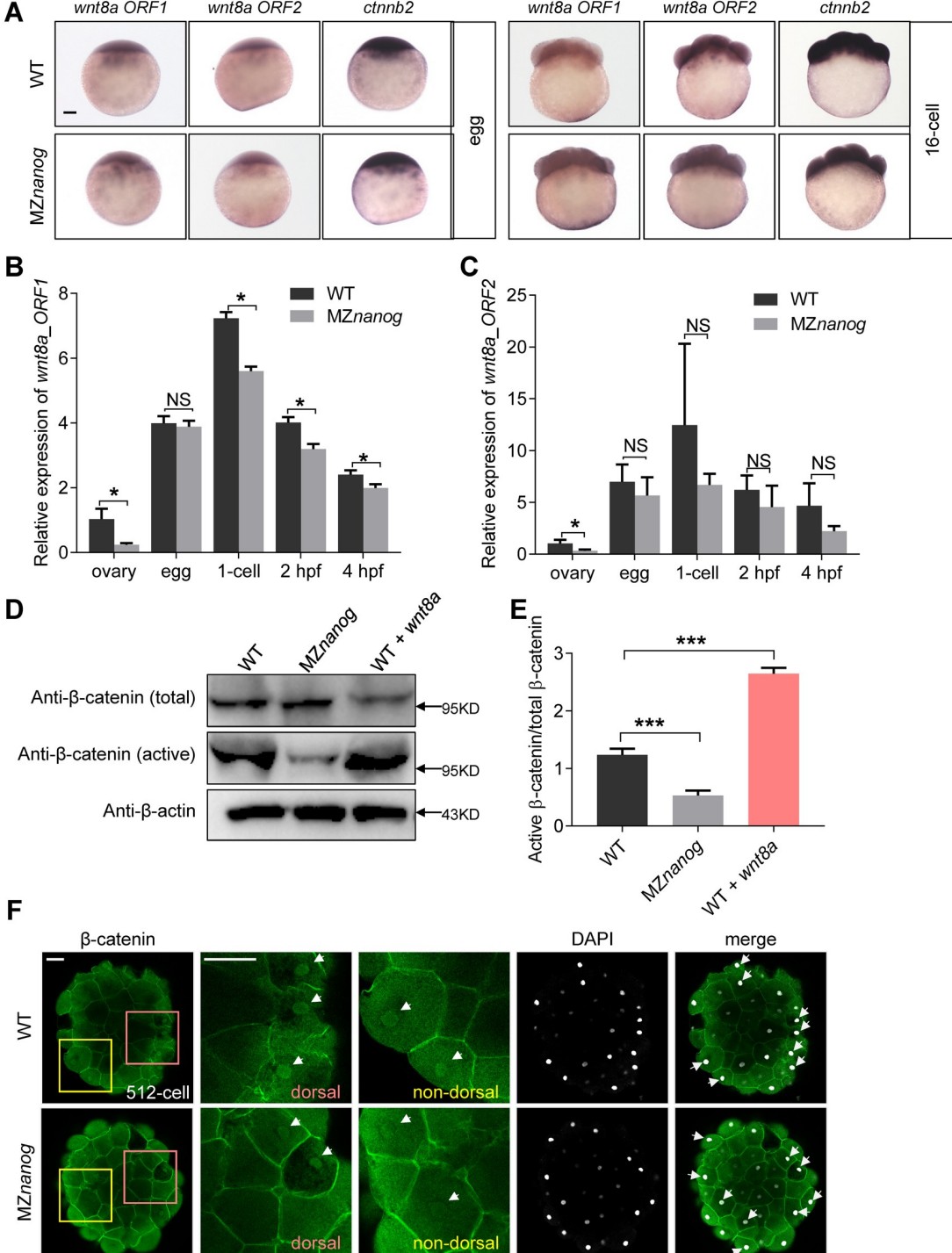

**Fig 5. Transcriptional level of Wnt/β-catenin components and nuclear accumulation of β-catenin are not increased in MZ*nanog*.** (A) WISH and (B, C) RT-qPCR showed the maternal transcription of *wnt8a1*, *wnt8a2*, and *ctnnb2* were not affected in MZ*nanog* eggs when compared with WT eggs. Scale bar, 100 μm. Error bars, mean ± SD, *$P < 0.05$; NS means no significant difference. (D) Western blot analysis of total β-catenin and nuclear β-catenin (active β-catenin) in WT, Mz*nanog*, and *wnt8a* overexpressed embryos. Anti-total β-catenin was used as the β-catenin expression control and anti-β-actin was used as the internal control. A total of 2 pg of *wnt8a* mRNA was injected at the 1-cell stage in WT and used as a positive control. Embryos were collected at 4 hpf. Experiments were carried out for triplicates. (E) Statistical analysis of active β-catenin/total β-catenin level in panel D. Error bars, mean ± SD, ***$P < 0.001$. (F) Immunolocalization of β-catenin on whole-mount embryos at the 512-cell stage shows that nuclear β-catenin in both the WT and MZ*nanog* was localized in dorsal margin cells and nondorsal

cells, and nuclear β-catenin localization was not stimulated in MZ*nanog* embryos. Signals were observed at animal view. Nuclei were co-stained with DAPI. Scale bar, 50 μm. The *P* values in this figure were calculated by Student *t* test. The underlying data in this figure can be found in S1 Data. hpf, hours post fertilization; MZ*nanog*, maternal zygotic mutant of *nanog*; RT-qPCR, reverse-transcription quantitative PCR; WISH, whole-mount in situ hybridization; WT, wild type.

which further supported the notion that Nanog functions as transcriptional activator during MZT.

We then checked the overall morphology of embryos in different rescue groups. Overexpression of either Nanog_FL or Nanog_ΔC could fully rescue the mutant phenotype, and the embryos could even survive to adulthood (Fig 6D–6F), indicating that the C-terminal of Nanog is not essentially required for its normal function. On the other hand, although overexpression of vp16-Nanog nearly rescued all the developmental defects of MZ*nanog*, the rescued embryo still showed a head-truncation phenotype (Fig 6D–6F), indicating that the elevated Wnt/β-catenin activity still persisted in the rescued embryos. To exclude the possibility that the head-truncation phenotype in vp16-*nanog* overexpressed embryos was caused by overexpression of Vp16, we generated a vp16-*nanog*_FL construct, which contains *vp16* and full-length *nanog*, for overexpression assay. Overexpression of vp16-*nanog*_FL could fully rescue the mutant phenotype, and the embryos could also survive to adulthood (S8 Fig). In addition, overexpression of the N-terminal deletion form of *nanog* (*nanog*_ΔN) could not rescue the development defect of MZ*nanog* (S8 Fig). These results demonstrate that the forebrain defect was not a side effect of the Vp16 fusion. Furthermore, in vitro TOPflash assay showed that co-transfection of *nanog*_FL or *nanog*_ΔC construct could efficiently suppress β-catenin-induced transcriptional activity, whereas co-transfection of vp16-*nanog* construct could not inhibit the activity (Fig 6G). All these results strongly suggest that the N terminal of Nanog is required for its suppressive activity on Wnt/β-catenin signaling.

## Nanog interferes with the binding of β-catenin to Tcf7

Since Nanog does not regulate the nuclear β-catenin level and its suppression of β-catenin transcriptional activity relies on its N terminal, we inquired whether Nanog physically interacts with β-catenin or its nuclear partners, such as activator-type TCF/Lef. Firstly, we identified which TCF is the activator-type TCF in activating maternal β-catenin activity in zebrafish. The expression of *tcf7* [71], *tcf7l2* (previous name, *tcf4*) [72], *tcf7l1a* (previous name, *tcf3a*) [73], and *tcf7l1b* (previous name, *tcf3b*) [64] were examined during oogenesis and early developmental stages. *tcf7* and *tcf7l2* showed strong maternal expression in developing oocytes and in pre-ZGA embryos. In contrast, *tcf7l1a* and *tcf7l1b*, which have shown to be repressor-type TCFs [64,73], displayed very weak expression at these stages (S9A and S9B Fig). We then focused on *tcf7* and *tcf7l2*, overexpression of *tcf7* or *tcf7l2* mRNA alone, or co-injection with *ctnnb2* mRNA, *tcf7* and *tcf7l2* could efficiently induce the ectopic expression of maternal β-catenin targets, *boz* and *chd* (S9C and S9D Fig). Moreover, overexpression of Tcf7 and Ctnnb2 could more efficiently induce the expression of *boz* and *chd* than overexpression of Tcf7l2 and Ctnnb2, suggesting that Tcf7 could serve as a strong activator-type TCF in mediating maternal β-catenin activity in zebrafish early development, which is in accordance with previous reports showing that Tcf7 acts as β-catenin-dependent *trans*-activators with Lef1 [71,74–76].

We then performed a co-immunoprecipitation (co-IP) assay between Nanog and Tcf7. Co-IP assay showed that Nanog physically interacts with Tcf7 (Fig 7A). To determine that which part of Nanog is capable of binding with Tcf7, we generated several deletion types of Nanog, including full-length *nanog* (myc-Nanog), N-terminal truncated Nanog (myc-Nanog-ΔN), homeodomain deleted Nanog (myc-Nanog-ΔH), C-terminal truncated Nanog (myc-Nanog-

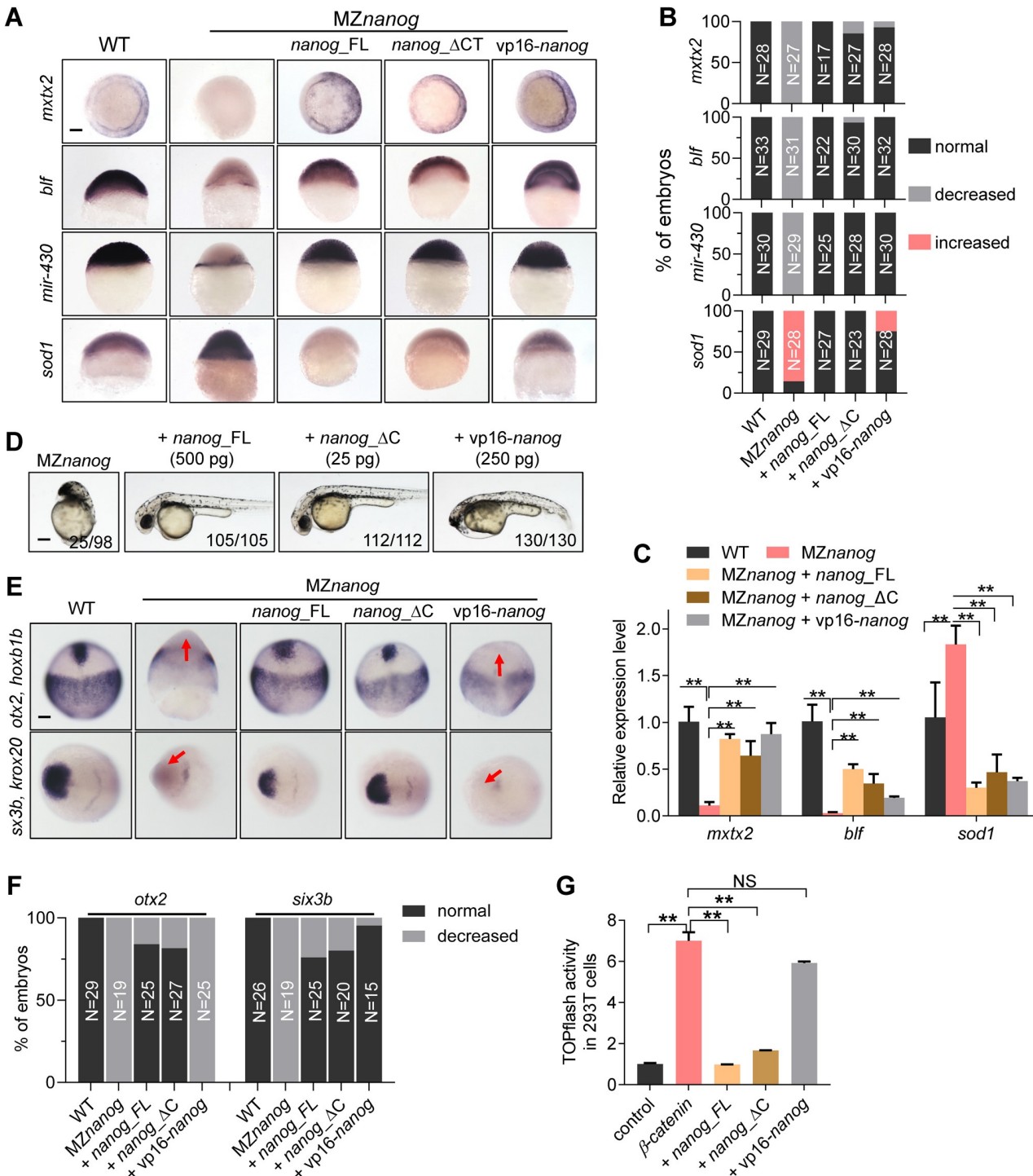

**Fig 6. N-terminal of Nanog is required for its Wnt/β-catenin repressive activity.** (A) WISH analysis showing the expression of mesendoderm marker, *mxtx2*, strictly zygotic gene, *blf*, and microRNA-430 precursor (*mir-430*), and miR-430 target, *sod1* in embryos of WT, MZ*nanog*, MZ*nanog* injected with *nanog*_FL, *nanog*_ΔC, or vp16-*nanog* mRNA. Expression of *mxtx2*, *blf*, and *mir-430* was reduced, even absent, whereas expression of *sod1* was significantly increased in MZ*nanog* embryos; overexpression of *nanog*_FL, *nanog*_ΔC, or vp16-*nanog* restored the expression of *mxtx2*, *blf*, and *mir-430* and cleaned the expression of *sod1* in MZ*nanog* embryos. *mxtx2*, *blf*, and *sod1* were detected at 6 hpf, and *mir-430* was detected at 4 hpf. Scale bar, 100 μm. (B) Statistical analysis of embryos in panel A. N represents analyzed embryo number. (C) Relative mRNA level of *mxtx2*, *blf*, and *sod1* in WT, M*znanog*, and the rescued embryos at 6 hpf examined by RT-qPCR analysis. Error bars, mean ± SD, **$P < 0.01$. (D) Overexpression of *nanog*_FL, *nanog*_ΔC, and vp16-*nanog* rescued the developmental defects of MZ*nanog*. Both of *nanog*_FL and *nanog*_ΔC rescued embryos showed WT-like phenotype, whereas vp16-*nanog* rescued embryos still showed a forebrain defective phenotype. Phenotype was observed at 36 hpf. Scale bar,

100 μm. The numbers below the morphology pictures indicate number of embryos showing representative phenotype/total number of embryos. (E) WISH analysis showing the expression of neuroectoderm marker *otx2* and forebrain marker *six3b* in embryos of WT, MZ*nanog*, MZ*nanog* injected with *nanog*_FL, *nanog*_ΔC, or vp16-*nanog* mRNA at 90% epiboly (for *otx2*) and 2-somite stage (for *six3b*). Expression of *otx2* and *six3b* was absent in MZ*nanog* embryos and restored by overexpression of *nanog*_FL or *nanog*_ΔC but not vp16-*nanog*. Red arrows indicate the absent expression of *otx2* or *six3b*. Scale bar, 100 μm. (F) Statistical analysis of the embryos in panel E. N represents analyzed embryo number. (G) TOPflash assay showing co-transfection of *nanog*_FL (2 μg) or *nanog*_ΔC (2 μg) but not vp16-*nanog* (2 μg) significantly inhibited the up-regulated β-catenin transcriptional activity induced by β-catenin (0.5 μg) in HEK293T cells. Error bars, mean ± SD, $^{**}P < 0.01$; NS means no significant difference. The *P* values in this figure were calculated by Student *t* test. The underlying data in this figure can be found in S1 Data. hpf, hours post fertilization; MZ*nanog*, maternal zygotic mutant of *nanog*; *nanog*_FL, full length of Nanog; *nanog*_ΔC, C-terminal truncated Nanog; RT-qPCR, reverse-transcription quantitative PCR; vp16-*nanog*, Nanog homeodomain fusion with Vp16; WISH, whole-mount in situ hybridization; WT, wild type.

ΔC), and N-terminal-only Nanog (myc-Nanog-NT). In a co-IP screening, we found that all types of Nanog proteins except Nanog-ΔN bind to Tcf7 (Fig 7A), suggesting the N-terminal of Nanog physically interacts with Tcf7.

Next, we aimed to understand which part of Tcf7 is responsible for its interaction with Nanog. We generated different forms of Tcf7 including full-length Tcf7, β-catenin binding domain-deleted form (myc-Tcf7_ΔN), Groucho/TLE binding domain (GroBD)-deleted form (myc-Tcf7_ΔGroBD), and the HMG deleted form (myc-Tcf7_ΔHMG). First, we confirmed the GroBD, β-catenin binding domain, and Lef1 binding domain of Tcf7. The full-length Tcf7 could efficiently bind to Gro2 (S10A Fig), β-catenin (S10B Fig), and Lef1 (S10C Fig), whereas the myc-Tcf7_ΔGroBD, myc-Tcf7_ΔN, and myc-Tcf7_ΔHMG could not bind to Gro2, β-catenin, and Lef1, respectively (S10A–S10C Fig). This result confirmed the respective binding domains of Gro2, β-catenin, and Lef1 on Tcf7. We then conducted a similar co-IP assay to learn the Nanog binding domain of Tcf7. As shown in Fig 7B, all forms of Tcf7 except the Tcf7_ΔGroBD could efficiently interact with Nanog, illustrating that Nanog and Groucho/TLE both have high binding affinity to the GroBD of Tcf7. Taken together, we have shown that the N-terminal region of Nanog physically interacts with the GroBD of Tcf7.

Because Nanog and β-catenin could both bind to Tcf7, there is a possibility that Nanog may interfere with the interaction between β-catenin and TCF in nucleus. We then performed a competitive binding assay in which the input of β-catenin and TCF was consistent in each sample and the input of Nanog was gradually increased. Strikingly, we found that the increased amount of Nanog effectively decreased the binding affinity of Tc7 to β-catenin (Fig 7C), indicating that Nanog may negatively regulate β-catenin transcriptional activity by attenuation of Tcf7/β-catenin transcriptional activator complex.

To challenge this conclusion, we co-transfected *ctnnbip1*, which can physically bind to β-catenin to prevent the association of β-catenin and TCF [35,77,78]. Interestingly, after co-transfected with *ctnnbip1*, increased amount of Nanog was co-immunoprecipitated with Tcf7 (Fig 7D), further supporting the conclusion that Nanog and β-catenin competitively interact with Tcf7. Finally, we verified the competitive binding with Tcf7 of β-catenin and Nanog in the MZ*nanog* mutant embryos. Myc-tcf7 was overexpressed in WT and MZ*nanog* embryos, and co-IP assay was conducted at 4 hpf. As expected, although the input of total β-catenin was somehow decreased in MZ*nanog* embryos compared with WT embryos, a significantly increased amount of endogenous β-catenin could be co-immunoprecipitated with Tcf7 in the absence of Nanog (Fig 7E and 7F). This confirms that Nanog interferes with the binding of β-catenin to Tcf7 in vivo.

## Interference of maternal β-catenin/Tcf7 transcriptional complex formation by Nanog safeguards proper dorsal development

The above data showed that Nanog binds to the Tcf7 and interferes with the interaction between β-catenin and Tcf7 in vitro and in vivo; we then tested this possibility using different

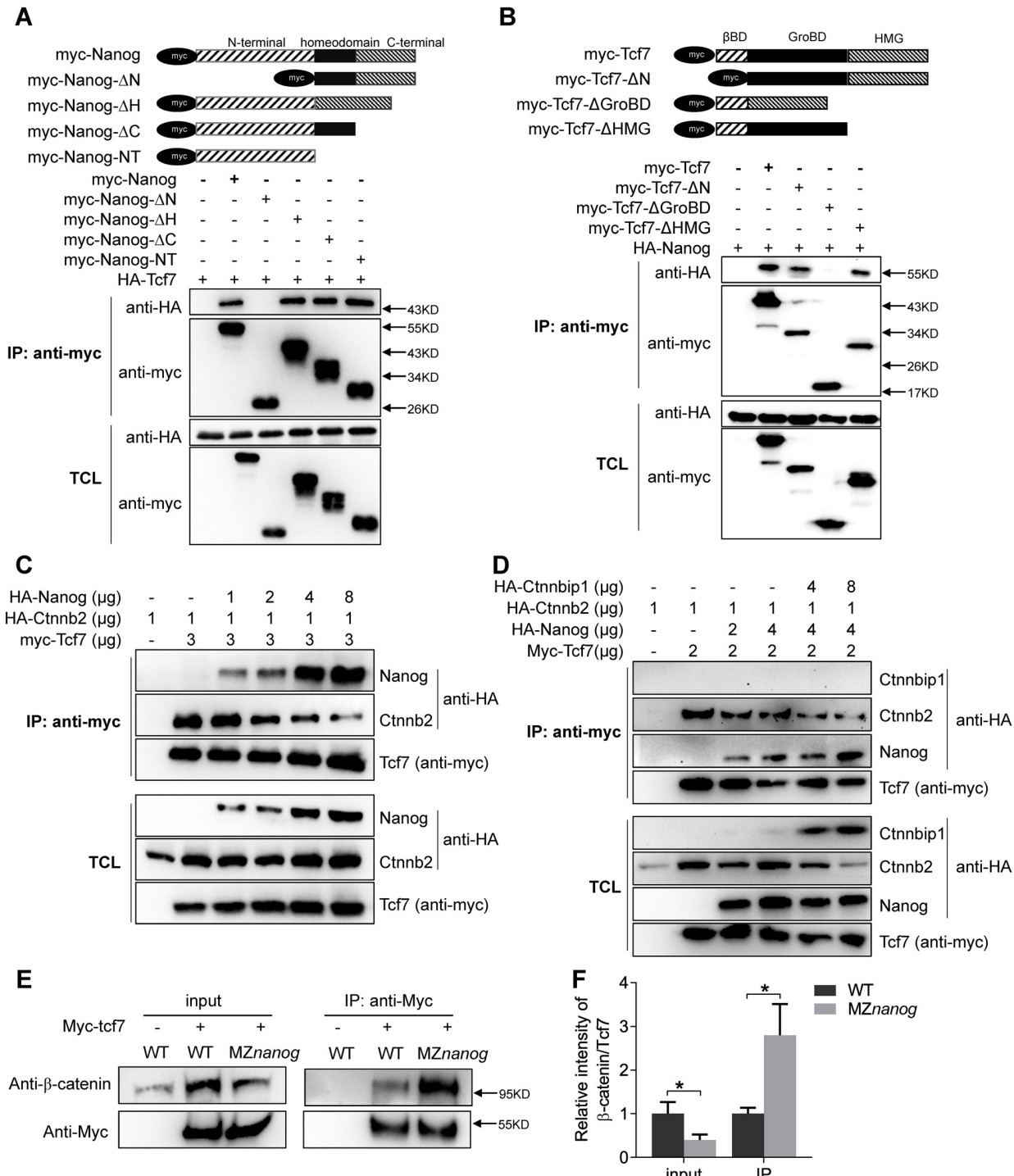

**Fig 7. Nanog interferes with the binding of β-catenin to Tcf7 in vitro and in vivo.** (A) Nanog interacts with Tcf7 through its N terminal. Different Myc-tagged Nanog were constructed and co-transfection with HA-Tcf7 in HEK293T cells. Among all the mutated types of Nanog, only the N-terminal truncated Nanog (Nanog-ΔN) could not coprecipitate with Tcf7, indicating that Nanog physically interacts with Tcf7 through its N terminal. (B) Tcf7 interacts with Nanog through its GroBD. Different Myc-tagged Tcf7 were constructed and co-transfection with HA-Nanog in HEK293T cells. Among all the mutated types of Tcf7, only the GroBD deleted Tcf7 (Tcf7_ΔGroBD) could not coprecipitate with Nanog, indicating that Tcf7 binds with Nanog through GroBD. (C) Nanog and β-catenin competitively binds with Tcf7. Co-transfection of increasing amount of Nanog decreases the interaction between β-catenin and Tcf7 in a dose-dependent manner. When increased amount of Nanog was transfected into Tcf7 and β-catenin co-transfected cells, decreased amount of β-catenin could be coprecipitated. The molecular weight of HA-Nanog is around 55 Kda, and HA-β-catenin is around 100 KDa, so we could distinguish the 2 anti-HA bands by different protein sizes. (D)

Co-transfection of Ctnnbip1 increased the binding affinity between Nanog and Tcf7. Two different amounts of HA-ctnnbip1 were co-transfected with Tcf7, Nanog and β-catenin in HEK293T cells; because Ctnnbip1 was overexpressed, an increased amount of Nanog was coprecipitated by myc-Tcf7. The molecular weight of HA-ctnnbip1 is around 10 KDa. Note that HA-β-catenin level was reduced when Ctnnbip1 was overexpressed. (E) Co-immunoprecipitation assay showed that increased amount of endogenous β-catenin could interact with Tcf7 in MZ*nanog* mutants compared with WT embryos. Embryos were collected at 4 hpf. (F) The intensity ratio of β-catenin to Tcf7 in panel E. Error bars, mean ± SD, *$P < 0.05$. The P values in this figure were calculated by Student t test. The underlying data in this figure can be found in S1 Data. GroBD, Groucho binding domain; HEK293T cells, human embryonic kidney 293T cells; MZ*nanog*, maternal -zygotic mutant of *nanog*; hpf, hours post fertilization; WT, wild type.

interfering molecules in MZ*nanog* embryos. We injected mRNAs encoding Tcf7_ΔβBD, Tcf7_ΔGroBD, Tcf7_ΔHMG, and Ctnnbip1 into MZ*nanog* at one-cell stage. Theoretically, Tcf7_ΔβBD can competitively bind to LEF1 without association of β-catenin, therefore resulting in decreased functional Tcf7/β-catenin/LEF1 complex. Similarly, Tcf7_ΔHMG can competitively bind with β-catenin but without the LEF1 association, therefore resulting in decreased level of functional Tcf7/β-catenin/LEF1 complex. Morphological analysis showed that overexpression of Tcf7_ΔβBD, Tcf7_ΔHMG, or Ctnnbip1 effectively rescued the developmental defects of MZ*nanog*, whereas overexpression of Tcf7_ΔGroBD did not induce any rescue (Fig 8A). WISH analysis of *chd* showed overexpression of Tcf7-ΔβBD, Tcf7_ΔHMG, or Ctnnbip1 effectively rescued the hyperdorsalization of MZ*nanog* embryos, which was characterized by a strong lateral and ventral extension of *chd* expression at 4.5 hpf (Fig 8B and 8C). These rescue effects were further confirmed by RT-qPCR analysis (Fig 8D). Finally, TOPflash assay proved that the elevated β-catenin transcriptional activity in MZ*nanog* embryos could be significantly reduced by overexpression of Tcf7-ΔβBD, Tcf7_ΔHMG, or Ctnnbip1 (Fig 8E).

All these results demonstrate that, Nanog and β-catenin competitively bind to Tcf7 and maintain the maternal β-catenin activity at homeostatic levels in different territories of WT embryo. In nondorsal cells, the high amount of maternally inherited Nanog binds to Tcf7, which safeguards Tcf7 from forming β-catenin/Tcf7 transcriptional activator complex with a low amount of nuclear β-catenin. In dorsal cells, however, the high amount of nuclear β-catenin competitively binds to Tcf7 even in the presence of nuclear Nanog to induce the transcription of dorsal genes, such as *boz* and *chd*. In contrast, in the absence of maternally provided Nanog in MZ*nanog* embryos, a low amount of nuclear β-catenin can form functional β-catenin/TCF complexes to activate the maternal β-catenin targets in the nondorsal cells, which in turn results in hyperdorsalization (Fig 8F).

## Discussion

The induction of vertebrate dorsal axis largely relies on nuclear accumulation of maternally provided β-catenin in the early embryo, which activates the dorsal genes, such as *boz*, *chd*, *sqt*, etc., in a cluster of dorsal cells [15]. These cells will develop into dorsal precursors, starting to establish embryonic dorsoventral axis. The significance of maternal β-catenin in establishing early dorsal axis is supported by gain- and loss-of-function studies in zebrafish; e.g., overexpression of β-catenin causes severe dorsalization and β-catenin-related mutant *ichabod* shows severe ventralization [25,79]. In the present study, we have detected a low amount of nuclear β-catenin accumulation in the nondorsal cells in zebrafish early embryo, which is consistent with the previous finding in *Xenpous* [56,57]. The nuclear accumulated β-catenin in nondorsal cells must be properly suppressed in order to prevent the embryo from ectopic activation of maternal β-catenin activity and hyperdorsalization. Through a series of genetic and biochemical studies, we uncovered a novel repressor of maternal β-catenin activity, Nanog, which suppresses the transcriptional activity of Wnt/β-catenin signaling by competitive binding to transcription-activator type Tcf7 and disrupting the functional β-catenin/TCF complex. Our

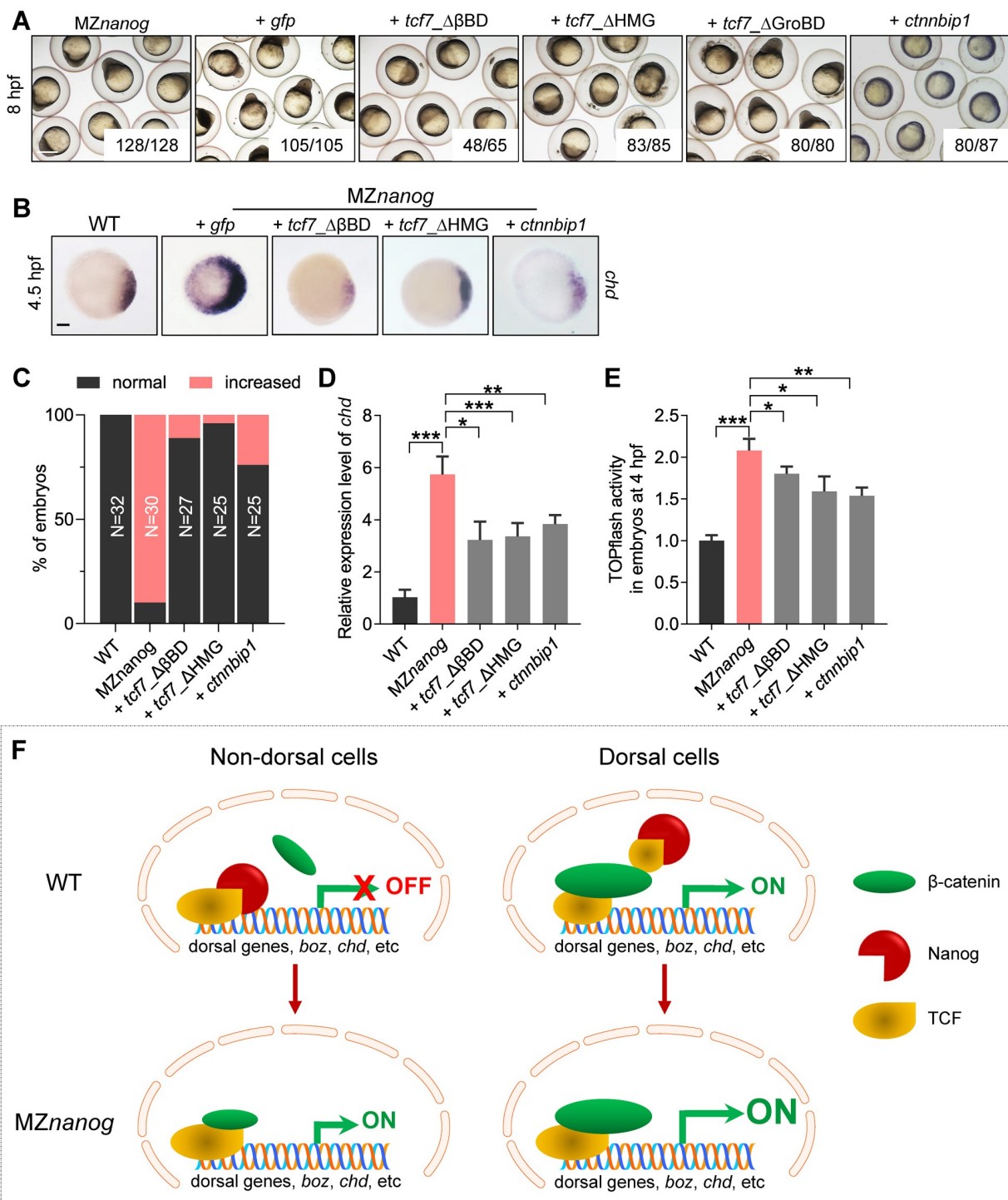

**Fig 8. Confrontation of the β-catenin transcriptional activity in nucleus rescues the developmental defect of MZ*nanog*.** (A) Injection of *tcf7*)_ΔβBD, *tcf7*_ΔHMG, or *ctnnbip1* mRNA rescued the early developmental defects of MZ*nanog*, whereas overexpression of *tcf7*_ΔGroBD did not. Phenotypes were observed at 8 hpf. At least 50 embryos were injected, and 3 independent experiments were performed. The numbers below the morphology pictures mean number of embryos showing representative phenotype/total number of embryos. Scale bar, 500 μm. (B) WISH analysis showing excessive and ectopic expression of *chd* in MZ*nanog* was rescued by overexpression of *tcf7*)_ΔβBD, *tcf7*_ΔHMG, or *ctnnbip1* at 4.5 hpf. Scale bar, 100 μm. (C) Statistical analysis of the embryos in panel B. N represents analyzed embryo number. (D) Relative mRNA level of *chd* in embryos of WT, MZ*nanog*, and MZ*nanog* injected with *tcf7*)_ΔβBD, *tcf7*_ΔHMG, or *ctnnbip1* mRNA at 4.5 hpf examined by RT-qPCR. Error bars, mean ± SD, $^*P < 0.05$, $^{**}P < 0.01$, $^{***}P < 0.001$. (E) Relative β-catenin transcriptional activity in embryos of WT, MZ*nanog*, and MZ*nanog* injected with *tcf7*)_ΔβBD, *tcf7*_ΔHMG, or *ctnnbip1* mRNA at 4 hpf examined by TOPflash assay. Error bars, mean ± SD, $^*P < 0.05$,

**P < 0.01, ***P < 0.001. (F) The model of Nanog repressing β-catenin transcriptional activity in nondorsal cell nuclei in WT embryo, and the ectopic activation of β-catenin transcriptional activity in the absence of Nanog in MZ*nanog* embryo. In nondorsal cells of WT embryo, the amount of nucleus-deposited maternal Nanog (red cartoon object) is much higher than that of the nuclear β-catenin (green cartoon object); therefore Nanog binds to TCF (yellow cartoon object), and the β-catenin transcriptional activity is not activated. In nondorsal cells of MZ*nanog* embryo, however, because of the absence of *nanog* in the nuclei, the small amount of nuclear β-catenin binds to TCF to activate the expression of dorsal genes (*boz*, *chd*, etc.), resulting in hyperdorsalization of the embryo. In dorsal cells of WT embryo, the amount of nuclear β-catenin is much higher than Nanog and facilitates the formation of β-catenin-TCF transcriptional complex to induce the expression of dorsal genes, *boz*, *chd*, etc. The *P* values in this figure were calculated by Student *t* test. The underlying data in this figure can be found in S1 Data. hpf, hours post fertilization; MZ*nanog*, maternal zygotic mutant of *nanog*; RT-qPCR, reverse-transcription quantitative PCR; *tcf7*)_ΔβBD, β-catenin-binding domain deleted Tcf7; *tcf7*_ΔHMG, high mobility group (LEF1-binding domain) deleted Tcf7; *tcf7*_ΔGroBD, Groucho-binding domain deleted Tcf7; WISH, whole-mount in situ hybridization; WT, wild type.

study therefore establishes the maternal Nanog, which has shown to play a central role in MZT [47,48] as a key factor that safeguards the embryo against global activation of maternal β-catenin activity.

In previous studies, maternal Nanog has shown to play a key role in controlling MZT, mainly including ZGA and the clearance of maternal mRNA, together with Pou5f1 and SoxB1 [47,48]. In our study, we generate MZ*nanog* mutants and show that the transcriptional activation of a series of zygotic genes is defective, and the clearance of maternal transcripts is strongly disrupted in MZ*nanog* embryos, strongly supporting the critical role of Nanog in mediating MZT. Recently, 2 independent studies by generating MZ*nanog* mutants report that maternal Nanog is also critical for extraembryonic tissue, embryo architecture, and cell viability [80,81]. In our study, we also observed those defects in the MZ*nanog* embryos. More importantly, we reveal that all these defects in MZ*nanog* embryos could be perfectly rescued by ubiquitous overexpression of full-length Nanog (Nanog_FL), C-terminal truncated Nanog (Nanog_ΔC), or a transcription-activator type Nanog (Vp16-Nanog). And even a relative LD of *nanog*_ΔC mRNA (25 pg/embryo) gave full rescue effect as the *nanog*_FL mRNA (500 pg/embryo). Therefore, our study clearly demonstrates that the C terminal of Nanog is not essentially required for mediating MZT and suppressing maternal β-catenin activity. Moreover, Nanog acts as a transcriptional activator in mediating MZT, mesendoderm formation, embryo architecture, and cell viability in zebrafish.

Previous in vitro studies have shown that Groucho/TLE is a major co-repressor that can replace β-catenin from the TCF/Lef complex in the absence of Wnt ligands, resulting in transcriptional repression of Wnt/β-catenin signaling pathway [10,12]. In the present study, we have carried out extensive knockout or knockdown studies with all the potential suppressive TLEs in zebrafish and conclude that all these TLEs do not likely suppress maternal β-catenin signaling during zebrafish early development. Although maternal Nanog is required for ZGA [47,48], the maternal β-catenin targets, such as *boz* and *chd*, are not transcriptionally silenced or down-regulated in MZ*nanog* embryos. Instead, they are hyperactivated in the absence of maternal Nanog. Therefore, although those maternal β-catenin targets are zygotic genes, their initial transcription is not triggered by Nanog but mainly activated by the maternal β-catenin activity that is repressed by maternal Nanog. Our study also shows that although the hyperactivated maternal β-catenin activity in MZ*nanog* is not due to the transcription up-regulation of *wnt8a* or *ctnnb2*, the hyperactivation of maternal β-catenin could be rescued by blocking maternally expressed Ctnnb2 activity. All these further highlight the significance of nucleus-located Nanog in regulating maternally provided nuclear β-catenin activity.

As a homeobox protein, zebrafish Nanog mainly functions as a transcriptional activator, just like Oct4 and Sox2 in humans [82]. In zebrafish MZ*nanog* embryo, although the Vp16-Nanog (N terminal of Nanog replaced by strong transcriptional activator Vp16) could perfectly substitute endogenous Nanog in the aspects of controlling MZT, it did not suppress

the elevated Wnt/β-catenin signaling activity in MZ*nanog* embryos. This indicates that the repression effect on Wnt/β-catenin signaling by Nanog is dependent on its N terminal but independent of its transcriptional activation activity. Our study reveals that the N terminal of Nanog could physically interact with the classical GroBD of Tcf7; this interaction interferes the formation of TCF/β-catenin transcriptional activation complex. This conclusion is supported by rescuing the dorsalization phenotype of MZ*nanog* by overexpression of Tcf7_ΔβBD, Tcf7_ΔHMG, or Ctnnbip1, which can disrupt the functional β-catenin/TCF complex. Therefore, our study has uncovered a novel role of Nanog in embryonic development: Nanog controls the maternal β-catenin transcriptional activity inside the nucleus to safeguard the embryo against global activation of maternal β-catenin and to form the primary dorsoventral axis. Given that the maternally inherited β-catenin has been considered as dorsal determinants in both fish and amphibian [15,16,83], the proper control of maternal β-catenin activity should be also crucial for dorsoventral axis formation in *Xenopus*. We have screened the *Xenopus* database but could not find any ortholog of Nanog in *Xenopus*. Nevertheless, a previous study has shown that Oct25, which is also required for the maintenance of pluripotency of ES cells, exhibit a similar function in suppression of maternal β-catenin activity in *Xenopus* [84].

In view of the key role of Nanog in activating the first wave of zygotic genes and clearance of maternal mRNA [47,48], our study further establishes a central role of Nanog in coordinating early embryonic development of zebrafish.

## Materials and methods

### Ethics statement

The experiments involving zebrafish followed the Zebrafish Usage Guidelines of the China Zebrafish Resource Center (CZRC) and were performed under the approval of the Institutional Animal Care and Use Committee of the Institute of Hydrobiology, Chinese Academy of Sciences under protocol number IHB2014-006.

### Zebrafish maintenance

All the zebrafish used in this study were maintained and raised as previously described [85] at the China Zebrafish Resource Center of the National Aquatic Biological Resource Center (CZRC-NABRC, Wuhan, China, http://zfish.cn). The WT embryos were collected by natural spawning from AB strain.

### Generation of *nanog* mutants by TALENs

The *nanog* mutants were generated by TALENs as previously described [66]. Two pairs of PCR primers listed in S1 Table were used to screen and distinguish *nanog* homozygous and heterozygous. The maternal zygotic *nanog* mutant line (MZ*nanog*) was derived and kept by injection of *nanog* mRNA at one-cell stage or derived from crossing of *nanog* female heterozygous with *nanog* male homozygous.

### Generation of *tle3a* and *tle3b* mutants by CRISPR/Cas9

The *tle3a* and *tle3b* mutants were generated using CRISPR/Cas9 as described previously [86]. The gRNA target and PAM sequence (underlined) of *tle3a* and *tle3b* are 5'-TGACAGAAAA GGCATAATCTGG-3' and 5'- AGCGAGCAGAGATATTTACCTGTGG-3'. pT3TS-zCas9 was used for Cas9 mRNA transcription; capped Cas9 mRNA was generated using T3 mMessage Machine kit (AM1344, Ambion, Austin, Texas). gRNA was generated using in vitro transcription by T7 RNA polymerase (P2075, Promega, Madison, Wisconsin). Cas9 mRNA and

gRNA were co-injected into WT embryos at one-cell stage. The primers used for mutant screening are listed in S1 Table.

## MO sequences and injections

MOs were obtained from Gene Tools (Philomath, Oregon). The sequence of the MOs used is: *nanog* MO: 5′-CTGGCATCTTCCAGTCCGCCATTTC-3′ (translation-blocking MO covering the translation initiation start, underlined). *wnt8a* MO1: 5′-ACGCAAAAATCTGGCAAGG GTTCAT-3′ [87], *wnt8a* MO2: 5′-GCCCAACGGAAGAAGTAAGCCATTA-3′ [87], *tcf7l1a* MO: 5′-CTCCGTTTAACTGAGGCATGTTGGC-3′ [64], *ctnnb1* MO: 5′-ATCAAGTCAGAC TGGGTAGCCATGA-3′ [88], *ctnnb2* MO: 5′-CCTTTAGCCTGAGCGACTTCCAAAC-3' [25], *tle3b* (gro1) MO: 5′-CGGCCCTGCGGATACATCTTGAATG-3′ [89], *tle3a* (*gro2*) MO: 5′-ATGTATCCTTTATTTATTGGAGCTC-3′ [90], *tle2a* MO: 5′-CATGGTGAATAGCGT GGTTTGTTGC-3′[91]. For all experiments, MO or combinations of MOs were injected in more than 50 embryos and experiments were reproduced at least 3 times.

Amount of MO injected in this study: *nanog* MO (LD): 0.5 ng/embryos; *nanog* MO (MD): 1.2 ng/embryo; *nanog* MO (no phenotype): 160 pg/embryo; *wnt8a*1 MO: 500 pg/embryo; *wnt8a2* MO: 500 pg/ embryo; *tcf7l1a* MO (head truncated): 1.6 ng/embryo; *tcf7l1a* MO (no phenotype): 800 pg/embryo; *ctnnb1* MO: 2 ng/embryo; *ctnnb2* MO: 2 ng/embryo; *tle3b* MO: 2 ng/embryo; *tle3a* MO: 2 ng/embryo; *tle2a* MO: 2 ng/embryo. A 1 μL sample was injected into approximately 1,000 embryos.

## mRNA synthesis and injection

For capped mRNA synthesis, full length, mutated, or truncated cDNAs were cloned into pCS2 + vector, subsequently linearized with *NotI* and transcribed using SP6 RNA polymerase using the mMESSAGE mMACHINE kit (AM1340, Ambion, Austin, Texas) and injected into one-cell stage embryos if not specified. pCS2+Nanog mismatch was constructed by mutating the *nanog* MO targeted sequence 5'-GAAATGGCGGACTGGAAGATGCCAG-3' to 5'-GATAT GGCAGATTGGAAAATGCCGG-3', and the amino acid sequence was not changed. All injection experiments were performed on more than 50 embryos and reproduced at least 3 times.

Amount of mRNA injected in this study: *wnt8a* mRNA (head truncated): 1 ng/μL (1 pg/ embryo); *wnt8a* mRNA (no phenotype): 0.1 ng/μL; *ctnnb2* mRNA: 200 ng/μL; *nanog*_FL mRNA: 500 ng/μL; *nanog* mismatch mRNA: 500 ng/μL; *nanog*_ΔC mRNA: 25 ng/μL; *nanog*_ΔN mRNA: 500 ng/μL; vp16-*nanog* mRNA: 250 ng/μL; vp16-*nanog*_FL mRNA: 250 ng/μL; En-*nanog* mRNA: 100 ng/μL; *tcf7* mRNA: 500 ng/μL; Myc-*tcf7* mRNA: 500 ng/μL; *tcf7l2* mRNA: 500 ng/μL; *hwa* mRNA: 200 ng/μL; *tcf7*_ΔβBD mRNA: 800 ng/μL; *tcf7*_ΔHMG mRNA: 800 ng/μL; and *ctnnbip1* mRNA: 400 ng/μL. GFP mRNA (100 ng/μL) was injected as control for rescue experiments.

For localized injection, dechorioned embryos were collected at the 32-cell stage and injected twice, one at margin cell and another at the middle cell of the blastula. *ctnnb2* mRNA or *hwa* mRNA were co-injected with *mCherry* mRNA, and *mCherry* was used as injection indicator.

For miR-430 mimics injection, a mixture of miR-430a, miR-430b, and miR-430c was injected at 3.3 μmol/μL. The miR-430 mimics and GFP-3xIPT-miR-430 reporter was injected as described [92]. A 1 μL sample was injected into approximately 1,000 embryos.

## In situ hybridization

PCR-amplified sequences of genes of interest were used as templates for the synthesis of an antisense RNA probe, labeled with digoxigenin-linked nucleotides. WISH on embryos were performed as described previously [93]. For in situ hybridization on the frozen section, adult

ovaries were stripped and fixed overnight with 4% PFA in PBS at 4 ˚C, then embedded with OCT and dissected at 10 μm. The procedures of hybridization followed a previous study [94].

## Immunofluorescence

Whole-mount immunofluorescence of β-catenin was carried out based on the standard protocol using mouse anti–β-catenin antibody (C7267, Sigma-Aldrich, Saint Louis, Missouri, 1:500 dilution). Different stages of embryos were fixed overnight with 4% PFA in PBS at 4 ˚C, washed with PBST (0.1% Triton X-100 added), and permeabilized with distilled water for 1 hour. The FITC conjugated goat anti-mouse antibody (F2761, Thermo, Waltham, Massachusetts, 1:500) was used as secondary antibody. After 4 times of washing in PBST with 0.1% Triton X-100, embryos were incubated in DAPI solution (5 μg/mL in PBST) for 1 hour at room temperature. Then, embryos were washed and mounted at animal view for observation. Immunostained embryos were scanned at Z-stack using a Leica SP8 confocal microscope as described previously [95].

For Nanog immunostaining, rabbit anti-Nanog polyclonal antibody was customized by ABclone (Wuhan, China) using a recombinant Nanog protein (1:200 dilution). Embryos were fixed overnight with 4% PFA in PBS at 4 ˚C and transferred into 30% sucrose/PBS, incubated at 4 ˚C for 1 day, mounted in mounting medium (4583, Tissue-Tek OCT Compound, Sakura, Torrance, California), and then cryosectioned. Cryosections of 10-μm thickness were used for immunostaining. Signals were photographed using a Leica SP8 confocal microscope as described previously [96].

## Luciferase assay

A total of 15 pg of TOPflash construct and 1.5 pg of Renilla reporter were mixed and co-injected into 1-cell stage embryos. A total of 4 ng of TOPflash construct and 0.4 ng of Renilla reporter were mixed and co-transfected into 293T cells with indicated plasmids. The injected embryos were collected at 4 hpf, and transfected cells were collected at 24 hours after transfection. The luciferase activity was measured by Dual-Luciferase Reporter Assay System (E1910, Promega, Madison, Wisconsin). TOPflash assays were performed in triplicate for each sample. Student $t$ test was used to assess the statistical significance.

## Stem-loop RT-PCR

Stem-loop RT-PCR was performed to quantify the expression of miR-430 as previously described [97]. Total RNAs were reversely transcribed using the miR-430-RT primer 5'-GTC GTATCCAGTGCAGGGTCCGAGGTATTCGCACTGGATACGACCTACCCCA-3' and the U6 RT primer 5'-AAAAATATGGAGCGCTTCACG-3'. The PCR primers were listed as follows: for U6, forward primer 5′-TTGGTCTGATCTGGCACATATAC-3′ and reverse primer 5′-AAAAATATGGAGCGCTTCACG-3′; for *miR-430a*, forward primer 5′- GCGAAGTGCTA TTTGTTGGGGT-3′ and reverse primer 5′-GTGCAGGGTCCGAGGT-3′, for *miR-430b*, forward primer 5′-GCGTGCTATCAAGTTGGGGTAG-3′ and reverse primer 5′-GTGCAGGGT CCGAGGT-3′.

## RT-qPCR

Relative abundance of target mRNAs was examined by RT-qPCR. Total RNAs were reversely transcribed using oligo dT, and PCR primers are listed in S1 Table. RT-qPCR was performed using the SYBRGreen Supermix (172–5124, BioRad, Hercules, California) on a BioRad CFX96.

## Immunoprecipitation and Western blot

For immunoprecipitation assays in culture cells, the different full length and truncated cDNAs were cloned into pCMV-myc (N-terminal myc tag) or in pCGN-HAM (N-terminal multiple HA tag) vectors. HEK293T cells were transiently transfected with the indicated constructs of interest using VigoFect (T001, Vigorous Biotechnology, Beijing, China); 24 hours after transfection, cells were harvested and lysed in RIPA buffer (50 mM Tris-HCl [pH 7.4], 150 mM NaCl, 1% NP40, 0.25% $C_{24}H_{39}O_4Na$, 1 mM EDTA, 1 mM NaF, and Protease Inhibitor Cocktail [S8830, Sigma-Aldrich]). Co-immunoprecipitation experiments were performed as described previously [61].

For in vivo immunoprecipitation assays, Myc-tcf7 was cloned into pCS2+ vector, and the mRNA was synthesized in vitro and injected into WT and MZ*nanog* embryos at one-cell stage. The injected embryos were collected at 4 hpf and lysed in RIPA solution. The noninjected WT embryo was used as negative control.

Primary antibodies and dilutions for western blot were Myc (sc-40, Santa Cruz Biotechnology, 1:2,000), HA (H3663, Sigma-Aldrich, 1:5,000), Nanog (ABclone, 1:2,000), total β-catenin (C7207, Sigma-Aldrich, 1:5,000), active β-catenin (8814, Cell Signal Technology, Danvers, Massachusetts, 1:1,000), β-actin (AC026, ABclone, 1:10,000). Signals were detected with ECL western blotting detection reagents (WBKLS0100, Millipore, Billerica, Manssachusetts) using ChemicDoc MP imaging system (BioRad).

## Statistical analysis

Significance of differences between means was analyzed using Student *t* test. Sample sizes were indicated in the figures or figure legends. Plotted mean was calculated by GraphPad software. Data were shown as mean ± SD. *P* value below 0.05 marked as *, *P* value below 0.01 marked as **, and *P* value below 0.001 marked as ***; NS means no significant difference.

## Supporting information

**S1 Fig. Nuclear β-catenin localizes in both dorsal and nondorsal cells.** Nuclear β-catenin distribution in zebrafish blastodermal cells was detected by whole-mount Immunofluorescence from 128-cell stage to high stage. Embryos were mounted, and the signal was detected at animal view. Nuclei were co-stained with DAPI. At least 15 embryos were detected in each stage. Arrow heads indicate the nuclear accumulation of β-catenin. Scale bar, 50 μm.
(TIF)

**S2 Fig. Localized injection of *ctnnb2* or *hwa* mRNA induce ectopic dorsal organizer.** (A) A diagram of localized injection into 2 cells at 32-cell stage. One injected cell is on the margin, and the other injected cell is located at the center of blastula. (B) *ctnnb2* or *hwa* mRNA was co-injected with *mCherry* mRNA and located injected mCherry fluorescence were observed at 4 hpf. Scale bar, 500 μm. (C) WISH analysis showing the ectopic expression of *boz* and *chd* induced by *ctnnb2* and *hwa* mRNA. Embryos with mCherry fluorescence were collected and examined by WISH. *boz* was detected at 4 hpf, and *chd* was detected at 4.5 hpf. The numbers below the WISH pictures are the number of embryos showing representative phenotype/total number of embryos. Scale bar, 100 μm. hpf, hours post fertilization; WISH, whole-mount in situ hybridization; WT, wild type.
(TIF)

**S3 Fig. Depletion of TLE did not elevate the maternal β-catenin activity.** (A) Functional domain analysis of zebrafish Tle2a, Tle2b, Tle2c, Tle3a, Tle3b, and Tle5. (B) The CRISPR/Cas9 target of *tle3a* is located within exon 6, and a 10-bp deletion mutant (*tle3a*^ihb355^) was obtained.

(C) The CRISPR/Cas9 target of *tle3b* is located at the splicing site of exon 2 and intron 2, and a 119-bp insertion mutant (*tle3b^{ihb354}*) was obtained. Target sequence is in bold, and the PAM sequence is in green. (D) Validation of *tle2a* MO effectiveness. *tle2a* MO target sequence was fused with GFP to result in pTle2a-GFP construct. pTle2a-GFP construct or pTle2a-GFP combination with *tle2a* MO was injected at one-cell stage. Fluorescence was observed at 10 hpf. Scale bar, 500 μm. (E) Knockdown of *tle2a* at 2 ng/embryo did not affect the early development of zebrafish. Scale bar, 500 μm. (F) Knockdown of *tle2a* (2 ng/embryo), *tle3a* (2 ng/embryo), and *tle3b* (2 ng/embryo), respectively, or in combination, does not lead to early embryonic development defect at 10 hpf and 56 hpf. MO or combined MOs was injected at one-cell stage, and at least 50 embryos were injected and observed. Scale bar, 500 μm. bp, base pair; GFP, green fluorescent protein; hpf, hours post fertilization; MO, morpholino; PAM, protospacer adjacent motif; WT, wild type.
(TIF)

**S4 Fig. *nanog* MO is specific.** (A) Three classes of phenotypes—WT-like, posterization, and dorsalization—were characterized in *nanog* morphants at 36 hpf. Scale bar, 100 μm. (B) Over-expression of *nanog* mismatched mRNA (*nanog* MO targeted site is mutated) rescued the posterization, and dorsalization defects of *nanog* morphants. N represents analyzed embryo number. The underlying data in this figure can be found in S1 Data. hpf, hours post fertilization; MO, morpholino; WT, wild type.
(TIF)

**S5 Fig. Characterization of *nanog* mutants.** (A) Generation of 2 *nanog* mutant alleles by TALEN. TALEN left and right arms are marked in green, and the target sequence is in red. Two different mutant lines were obtained: 2-bp deletion line (*nanog^{ihb97}*) and 1-bp insertion line (*nanog^{ihb98}*). The 2-bp deletion leads to frame-shift of Nanog protein, and the 1-bp insertion results in premature termination of Nanog protein at the mutation site. (B) Phenotype characterization of M*nanog^{ihb97}*, Z*nanog^{ihb97}*, and 2 different mutants, MZ*nanog^{ihb97}* and MZ*nanog^{ihb98}*. Both of M*nanog^{ihb97}* and two types of maternal -zygotic *nanog* mutants, MZ*nanog^{ihb97}*, MZ*nanog^{ihb98}* show slow development and abnormal cell movement, then die within 24 hpf. However, Z*nanog^{ihb97}* mutant shows normal development and reproduction, the same as WT. Scale bar, 100 μm. (C) Maternal *nanog* expression disappeared and small amount of zygotic *nanog* was detected in MZ*nanog^{ihb97}*. Both low expression of maternal and zygotic *nanog* were detected in MZ*nanog^{ihb98}*. Scale bar, 100 μm. bp, base pair; hpf, hours post fertilization; M*nanog*, maternal mutant of *nanog*; MZ*nanog*, maternal -zygotic mutant of *nanog*; TALEN, transcription activator-like effector nuclease; WT, wild type.
(TIF)

**S6 Fig. Decreased BMP activity in MZ*nanog* is rescued by knocking down of *ctnnb2*.** (A) Relative mRNA level of *bmp2b*, *bmp7*, and BMP target *vent* were reduced in MZ*nanog* examined by RT-qPCR analysis. Depletion of *ctnnb2* could restore the abnormal expression of *bmp2b*, *bmp7*, and *vent*, whereas knockdown of *ctnnb1* could not. *bmp2b*, *bmp7*, and *vent* was detected at 6 hpf. β1 MO, *ctnnb1* MO; β2 MO, *ctnnb2* MO. Error bars, mean ± SD, $^{**}P < 0.01$; NS means no significant difference. (B) Relative mRNA level of *admp* was significantly decreased in MZ*nanog* at 6 hpf, and *radar* was significantly up-regulated at 2 hpf by RT-qPCR analysis. Error bars, mean ± SD, $^{***}P < 0.001$. The *P* values in this figure were calculated by Student *t* test. The underlying data in this figure can be found in S1 Data. hpf, hours post fertilization; MO, morpholino; MZ*nanog*, maternal zygotic mutant of *nanog*; RT-qPCR, reverse-transcription quantitative PCR; WT, wild type.
(TIF)

**S7 Fig. Vp16-Nanog could fully rescue the MZT defects in MZ*nanog*.** (A) The diagram of Vp16-Nanog (vp16-Nanog homeodomain) and En-Nanog (Enrgrailed2-Nanog homeodomain). The transcription activator VP16 or repressor Enrgrailed2 was fusion with Nanog homeodomain to result in Vp16-Nanog or En-Nanog. (B) WISH analysis showing the expression of mesendoderm marker, *mxtx2*, strictly zygotic gene, *blf*, and microRNA-430 precursor (*mir-430*), and miR-430 target, *sod1* in embryos of WT, WT injected with *nanog* MO, En-*nanog* or vp16-*nanog*. Expression of *mxtx2*, *blf*, and *mir-430* was reduced, even absent, in *nanog* morphants and En-Nanog injected embryos but enhanced in vp16-*nanog* overexpressed embryos. The expression of miR-430 target gene, *sod1*, failed to be cleared in *nanog* morphants and En-Nanog injected embryos at shield stage. Scale bar, 100 μm. (C) Statistical analysis of embryos in panel B. N represents analyzed embryo number. (D) The fluoresce intensity of GFP-3xIPT-miR-430 reporter, which carries a target sequence of miR-430, is negative correlated with the expression of miR-430. The intensity of GFP was higher in MZ*nanog* than WT at 4 hpf and 6 hpf, indicating deletion of *nanog* resulted in inactivation of miR-430 expression. Meanwhile, overexpression of miR-430 mimics, *nanog*, or vp16-*nanog* in MZ*nanog* restored the high expression of GFP-3xIPT-miR-430 reporter. Scale bar, 500 μm. (E and F) Relative expression level of miR-430a (E) and miR-430b (F) were inactivated in MZ*nanog* examined by stem-loop PCR; overexpression of *nanog*_FL or vp16-*nanog* can fully restore the expression failure of miR-430a and miR-430b at 4 hpf, 6 hpf, and 8 hpf. Error bars, mean ± SD, $^*P < 0.05$, $^{**}P < 0.01$, $^{***}P < 0.001$. The *P* values in this figure were calculated by Student *t* test. The underlying data in this figure can be found in S1 Data. hpf, hours post fertilization; MZ*nanog*, maternal -zygotic mutant of *nanog*; MZT, maternal zygotic transition; RT-qPCR, reverse-transcription quantitative PCR; WISH, whole-mount in situ hybridization; WT, wild type. (TIF)

**S8 Fig. Nanog N terminal is required for repression of maternal β-catenin activity.** Overexpression of *nanog*_FL and vp16-*nanog*_FL could fully rescue the developmental defect of MZ*nanog*, and N-terminal truncated *nanog* (*nanog*_ΔN) could not. The numbers below the embryo pictures are the number of embryos showing representative phenotype/total number of embryos. Scale bar, 100 μm. MZ*nanog*, maternal zygotic mutant of *nanog*; *nanog*_FL, full length of Nanog; *nanog*_ΔN, N' terminal truncated Nanog; vp16-*nanog*_FL, full length of Nanog fusion with Vp16. (TIF)

**S9 Fig. Tcf7 is a strong activator-type TCF in zebrafish.** (A) Maternal expression of *tcf7*, *tcf7l2*, *tcf7l1a*, and *tcf7l1b* was detected by in situ hybridization on cryosections of ovaries. Scale bar, 100 μm. (B) WISH analysis showing *tcf7*, *tcf7l2*, *tcf7l1a*, and *tcf7l1b* were maternally deposited at unfertilized egg, 2-cell stage, and 2 hpf embryos. Scale bar, 100 μm. (C) Injection of low dose of *ctnnb2* mRNA (200 pg/embryo), *tcf7* mRNA, or *tcf7l2* mRNA alone, or co-injection of low dose of *ctnnb2* mRNA (200 pg/embryo) with *tcf7*, or *tcf7l2* at 1-cell stage and detection the expression of *boz* and *chd* by WISH. Both individual injection and co-injection of TCF with *ctnnb2* mRNA induced up-regulation of *boz* and *chd*, and Tcf7 coordination with Ctnnb2 showed more efficient induction of *boz* and *chd* than Tcf7l2 and Ctnnb2, suggesting Tcf7 could serve as a strong activator-type TCF in mediating maternal β-catenin activity. *boz* was detected at 4 hpf; *chd* was detected at 4.5 hpf. Scale bar, 100 μm. (D) Statistical analysis of the embryos in panel C. N represents analyzed embryo number. The underlying data in this figure can be found in S1 Data. hpf, hours post fertilization; TCF, T-cell factor; WISH, whole-mount in situ hybridization; WT, wild type. (TIF)

**S10 Fig. Interaction of Tcf7 with Groucho/TLE, β-catenin, and LEF1.** (A) Tcf7 interacts with Groucho/TLE through predicted GroBD. Different Myc-tagged Tcf7 were constructed and co-transfection with HA-Groucho2 in HEK293T cells. Deletion of predicted GroBD of Tcf7 disrupts its interaction with Groucho2, indicating that Groucho2 physically binds to the potential GroBD of Tcf7. (B) Tcf7 interacts with β-catenin through its N terminal. Deletion of the N terminal of Tcf7 disrupts its interaction with β-catenin, indicating that β-catenin physically interacts with the N terminal of Tcf7. (C) Tcf7 interacts with LEF1 through its HMG domain (C terminal). Deletion of HMG domain of Tcf7 disrupts its interaction with Lef1, indicating that Lef1 physically interacts with the HMG domain of Tcf7. GroBD, Groucho/TLE binding domain; HEK293T, human embryonic kidney 293T; HMG, high mobility group; TLE, transducin-like enhancer of split.
(TIF)

**S1 Raw Images. The uncropped blots for the westerns.**
(PDF)

**S1 Table. The primers for RT-qPCR and mutant screening.** RT-qPCR, reverse-transcription quantitative PCR.
(DOCX)

**S1 Data. Numerical data used in Figs 1, 2, 3, 4, 5, 6, 7 and 8 and S4, S6, S7 and S9 Figs.**
(XLSX)

## Acknowledgments

We thank Kuoyu Li from the China Zebrafish Resource Center of the National Aquatic Biological Resource Center (CZRC-NABRC) for zebrafish raising and management. We thank Fang Zhou from Analysis and Testing Center of Institute of Hydrobiology, CAS for technique support of confocal imaging, Dr. Yun Liu from Sun Yat-sen University for denoting miR430 mimics and GFP-3xIPT-miR-430 reporter, Prof. Anming Meng from Tsinghua University for providing pCS2+*hwa* construct.

## Author Contributions

**Conceptualization:** Yonghua Sun.

**Data curation:** Mudan He, Yonghua Sun.

**Funding acquisition:** Yonghua Sun.

**Investigation:** Ru Zhang, Shengbo Jiao, Fenghua Zhang.

**Methodology:** Mudan He, Ru Zhang.

**Project administration:** Yonghua Sun.

**Resources:** Ding Ye, Houpeng Wang.

**Supervision:** Yonghua Sun.

**Validation:** Mudan He, Ru Zhang, Shengbo Jiao, Fenghua Zhang.

**Writing – original draft:** Mudan He.

**Writing – review & editing:** Yonghua Sun.

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
