## [Editor Report · Decision Letter 0]

23 Oct 2019

Dear Dr Sun, 

Thank you for submitting your manuscript entitled "Nanog safeguards early embryogenesis against global activation of maternal β-catenin activity by interfering with TCF factors" for consideration as a Research Article by PLOS Biology.

Your manuscript has now been evaluated by the PLOS Biology editorial staff as well as by an academic editor with relevant expertise and I am writing to let you know that we would like to send your submission out for external peer review.

Please re-submit your manuscript within two working days, i.e. by Oct 25 2019 11:59PM.

Kind regards,

Di Jiang

PLOS Biology

---

## [Decision Letter · Decision Letter 1]

12 Nov 2019

Dear Dr Sun,

Thank you very much for submitting your manuscript "Nanog safeguards early embryogenesis against global activation of maternal β-catenin activity by interfering with TCF factors" for consideration as a Research Article at PLOS Biology. Your manuscript has been evaluated by the PLOS Biology editors, an academic editor with relevant expertise, and by three independent reviewers.

The reviews of your manuscript are appended below. You will see that the reviewers find the work potentially interesting. However, based on their specific comments and following discussion with the academic editor, I regret that we cannot accept the current version of the manuscript for publication. We would be willing to consider resubmission of a comprehensively revised version that thoroughly addresses all the reviewers' comments. We particularly want to see that reviewer 2's concerns are conscientiously addressed. We cannot make any decision about publication until we have seen the revised manuscript and your response to the reviewers' comments. Your revised manuscript would be sent for further evaluation by the reviewers.

We appreciate that these requests represent a great deal of extra work, and we are willing to relax our standard revision time to allow you six months to revise your manuscript. Please email us (plosbiology@plos.org) to discuss this if you have any questions or concerns, or think that you would need longer than this. At this stage, your manuscript remains formally under active consideration at our journal; please notify us by email if you do not wish to submit a revision and instead wish to pursue publication elsewhere, so that we may end consideration of the manuscript at PLOS Biology.

Your revisions should address the specific points made by each reviewer. Please submit a file detailing your responses to the editorial requests and a point-by-point response to all of the reviewers' comments that indicates the changes you have made to the manuscript. In addition to a clean copy of the manuscript, please upload a 'track-changes' version of your manuscript that specifies the edits made. This should be uploaded as a "Related" file type. You should also cite any additional relevant literature that has been published since the original submission and mention any additional citations in your response. 

Before you revise your manuscript, please review the following PLOS policy and formatting requirements checklist PDF: http://journals.plos.org/plosbiology/s/file?id=9411/plos-biology-formatting-checklist.pdf. It is helpful if you format your revision according to our requirements - should your paper subsequently be accepted, this will save time at the acceptance stage.

Please note that as a condition of publication PLOS' data policy (http://journals.plos.org/plosbiology/s/data-availability) requires that you make available all data used to draw the conclusions arrived at in your manuscript. If you have not already done so, you must include any data used in your manuscript either in appropriate repositories, within the body of the manuscript, or as supporting information (N.B. this includes any numerical values that were used to generate graphs, histograms etc.). For an example see here: http://www.plosbiology.org/article/info%3Adoi%2F10.1371%2Fjournal.pbio.1001908#s5.

For manuscripts submitted on or after 1st July 2019, we require the original, uncropped and minimally adjusted images supporting all blot and gel results reported in an article's figures or Supporting Information files. We will require these files before a manuscript can be accepted so please prepare them now, if you have not already uploaded them. Please carefully read our guidelines for how to prepare and upload this data: https://journals.plos.org/plosbiology/s/figures#loc-blot-and-gel-reporting-requirements.

Upon resubmission, the editors will assess your revision and if the editors and Academic Editor feel that the revised manuscript remains appropriate for the journal, we will send the manuscript for re-review. We aim to consult the same Academic Editor and reviewers for revised manuscripts but may consult others if needed.

If you still intend to submit a revised version of your manuscript, please go to https://www.editorialmanager.com/pbiology/ and log in as an Author. Click the link labelled 'Submissions Needing Revision' where you will find your submission record. 

Sincerely,

Di Jiang

PLOS Biology

Reviewer remarks:

Reviewer #1: 

Previous studies in zebrafish have established that maternal Nanog plays a role in clearing maternal mRNAs for midblastula transition and is also required for extraembryonic tissue development and cell variability. This manuscript investigated maternal nanog functions using maternal/zygotic nanog mutants and through biochemical experiments. It first observed that both Mnanog and MZnanog mutants exhibited dorsalized phenotypes, and then established that the dorsalized phenotype was associated with over-activation of beta-catenin signaling before midblastula transition. The authors finally uncovered the novel mechanism by which maternal Nanog binds to beta-catenin to prevent the latter from binding to the transcriptional activator TCF7/4. This study is comprehensive and the major conclusions are well supported by experimental data. The manuscript is well organized. To improve the manuscript, the following issues needed to be considered. 

Major concerns:

1) It is now well known that cytoplasmic beta-catenin can be stabilized and translocated into the nucleus in various ways independent of Wnt ligands/Wnt receptors-mediated signaling. Thus, beta-catenin signaling is not necessarily identical to Wnt signaling. In Xenopus and zebrafish, increasing evidence suggest that Wnt ligands/Wnt receptors signaling is not activated before midblastula transition. The authors should be cautious about this issue. For example, the authors in several context mentioned hyperactivation/overactivation of maternal Wnt activity in MZnanog, and this is inappropriate. 

2) Fig 4A showed that Nanog protein level at the shield stage was sharply dropped compared to that at the sphere stage, but Fig. 2C showed Nanog protein level at 6 hpf (shield stage) was much higher than that at 4 hpf. This inconsistency should be confirmed or explained. 

3) The dynamics of beta-catenin in nuclei of non-dorsal cells should be examined. Fig. 5f shows no nuclear beta-catenin accumulation in the ventral part at 3 hpf, but its accumulation in dorsal blastomeres is apparent at 128-cell stages. How does this difference arise?

4) The authors claimed that Nanog could suppress maternal beta-catenin in non-dorsal cells. Given that Nanog is maternally expressed in all blastomeres, how is its inhibitory role in dorsal blastomeres prevented?

5) The authors observed dorsalization, posteriorization and MZT defects in nanog morphants and MZnanog, mRNA rescue experiments should be performed to confirm the specific phenotypes, especially in the morphants. And Real-time PCR should be performed to further confirm ISH results in Figs. 2g, 2i, 2l, 3c, 6a, 6d and 8b.

6) The authors propose a competition mechanism between Nanog and beta-catenin for their binding with TCF7. In Fig.7c, they show decreased binding of β-catenin and TCF7 when Nanog is simultaneously transfected. But in the lanes 2-4 of Fig. 7d, beta-catenin/TCF7 complex was increased in the presence of Nanog compare to the case in absence of Nanog. This experiment should be repeated. In addition, the authors should check TCF7/β-catenin interaction after knockdown of nanog or in MZnanog mutants to further confirm this competition mechanism.

7) Results in Fig. 5d and Fig. 5e are not consistent. The lane 2 in Fig. 5d showed obvious reduction of active beta-catenin level, while the statistic result in Fig. 5e showed no significant change. This experiment needed biological repeats. 

Minor concerns:

1) Scale bars should be added in embryos pictures.

2) Legends for Fig. 3g is missing.

3) Protein markers should be labeled for WB results in Fig. 2c, Fig. 4a, Fig. 7 and Fig. S5.

4) Pictures in Fig. 2g (embryo stage for boz in MO group is not consistent); Fig. 4c and Fig. 6a should be improved.

5) ENH mRNA in Line 341 should be substituted by EN-Nanog.

6) Grammatical mistakes in Line 46/88/257/518 should be corrected.

7) Double check typos. For example, line 270, “maternal deposit” should be “maternally deposited”; line 278, “was completed” should be “was completely”; line 462, “serve ventralization” should be “severe ventralizatio

Reviewer #2: 

The manuscript by He et al investigates the interactions between beta-catenin signaling and homeodomain protein Nanog in zebrafish. The novel results in this paper are mostly coming from in-vitro and overexpression assays; in-vivo Nanog loss-of function data are largely redundant with the previous publications (cited in the paper). Using beta-catenin reporter plasmid (Topflash –luciferase reporter assay), the authors show that the beta-catenin signaling levels are elevated in zebrafish embryos upon Nanog loss-of function. Using overexpression assays in zebrafish embryos and 293 HEK cells, the authors further show that Nanog antagonizes beta-catenin-dependent activation of the reporter plasmid. They further demonstrate by co-immunoprecipitation, that N-terminal domain of Nanog is responsible for in-vitro binding to Tcf7, and may block beta-catenin-TCF7 interaction. However, from my point of view, the authors did not provide enough evidence that this in-vitro mechanism is somewhat important for vertebrate development. The global conclusions of the manuscript, claimed already in the title “Nanog safeguards early embryogenesis against global activation of maternal ß-catenin activity by interfering with TCF factors” are not at all validated by the presented results (see the major points below). The manuscript, in its present form, cannot be published without additional extensive experimental work, which may challenge the data interpretation. Alternatively, the novel in-vitro part can be published separately in a more specialized journal. 

Major comment 1:

In the introduction, the authors completely ignore a solid part of published zebrafish research, which does not fit to their conclusions. Namely, it is firmly established that maternal and zygotic beta-catenin signaling acts in the opposing directions: zygotic Wnt represses the organizer, acting in parallel with BMP signaling. The zebrafish genome contains two β-catenin genes, β-catenin-2 and β-catenin-1. The formation of the dorsal organizing center depends on maternal expression of β-catenin-2, while both β-catenins repress the organizer and promote ventral development zygotically (Bellipanni et al. 2006). Bozozok/dharma and Chordin are repressed by zygotic Wnt through its targets, ventral transcriptional repressors Vox, Vent and Ved (Melby et al. 1997; Kawahara and Dawid 2000; Ramel and Lekven 2004; Ramel et al. 2005; Varga et al. 2007). In Maternal-Zygotic Nanog mutants, Vox, Vent as well as BMP2 are strongly downregulated, which is documented in three publications, cited by authors (Fig.7 in ref.42; Fig.3 A,B in ref. 41; Fig.4 in ref.52). In the He et al. manuscript, increase in chd and boz in Nanog mutants or morphants is iteratively referred to as a single in-vivo evidence that maternal beta-catenin signaling levels are increased upon Nanog loss-of function (lane 201, 208,209, 250, 289,301,Fig.2g,h, Fig.4 c,d,g,e, discussion). The alternative and sufficient explanation for increase in the organizer genes and for the dorsalized phenotype of the Nanog mutants is a documented decrease in zygotic b-cat activity targets and BMPs. This explanation is directly the opposite to what the authors conclude from their experiments, it should be discussed and should not be simply ignored.

Major comment 2:

Related to comment 1: In-vitro, the authors measured endogenous beta-catenin activity using beta-catenin-responsive Topflash plasmid (TCF1 binding sites fused to the luciferase reporter gene). The authors show that beta-catenin- induced activity of luciferase in zebrafish embryos (Fig.3e, f,g; Fig.4f). It is absolutely unclear to me, why the authors interpret their results as an increase of maternal beta-catenin activity (lane 250 on): DNA, injected into the early embryo, can be only transcribed after ZGA, the luciferase is measured at 8 hpf, late gastrula stage. At best, the authors measure some complex mixture of maternal and zygotic activities of both beta-catenins (provided that luciferase half-life is 3-4 hours). The fact, that there is more unbound nuclear beta-catenin in MZnanog, can be interpreted in two different ways. The authors suggest, that Nanog represses the binding of maternal beta-catenin to TCFs by means of N-terminus, blocking DNA-bound TCFs. Alternative interpretation, for the embryos, compatible with zygotic repression of Vent Vox and Ved genes in MZnanog (refs 41,42,52)- would be that in the ventral cells of MZnanog, zygotic beta-catenin cannot efficiently access its targets on DNA without Nanog pioneering activity, so that excess of beta-cat remains in the nucleus and activates the ectopic plasmid. In fact, considering the results in 293 HEK cells, both explanations can be true. This issue has to be discussed. 

Major comment 3:

Figure 6C, lane 330 – on. The authors claim that N-terminus of Nanog is required for supression of beta-catenin activity, based on the MZnanog rescue experiments. They show, that all gastrulation defects in MZnanog can be completely rescued by the full-length protein, c-terminal-truncated Nanog, and by N-terminal truncated Nanog, fused to VP-16 transcriptional activating domain (Nanog-VP16). However, the first two constructs rescue the embryos to full viability, while the injection of N-terminal truncated Nanog, fused to VP-16 results in posteriorized embryos with forebrain defects. The authors claim that forebrain defects result from the absence of Nanog N-terminus in Nanog-VP16. Alternatively, the forebrain defects may be an effect of VP-16 fusion. Two additional experimental controls are absolutely necessary to distinguish between the possibilities: rescuing MZnanog with 1) full-length Nanog – VP16 fusion and 2) N-terminally truncated Nanog. If N-terminus of Nanog is required for embryo development, full-length-VP16 would completely rescue the MZnanog embryos, and N-terminal truncation will produce the embryos with forebrain defects (or it would not rescue at all). If otherwise, the in-vitro interaction between TCF7 and Nanog is dispensable for development. 

Major comment 4:

(first chapter of the results, line 133 on). Groucho is known transcriptional repressor of beta-catenin activity, first found in Drosophila. Vertebrate genomes contain multiple homologs of Groucho: for instance, Zebrafish genome has 6 homologs: tle2a, tle2b, tle2c, tle3a, tle3b and tle5 (zfin.org). The authors knocked down two out of 6 zebrafish Grouchos, tle3a and tle3b. They do not see the developmental defects in single and double mutants, which allows them to conclude that: “Tle3a and Tle3b do not contribute to the repression of maternal beta-catenin activity”. To my opinion, the conclusion is misleading: is well possible that Tle3a and Tle3b contribute redundantly to the repression of the maternal b-cat 2 activity together with the other maternal Grouchos. At least one Groucho homolog, tle2a is expressed maternally in a high level, even exceeding tle3a and tle3b (refs: Thisse 2001, zfin.org, and “A high-resolution mRNA expression time course of embryonic development in zebrafish”, White et al., Elife, 2017). To substantiate and validate their statement, the authors should knock down at three Grouchos (i.e. by injecting tle2a Moropholino to the double MZtle3a/b mutant); and show that the other members (tle2b, tle2c and tle5) are not expressed maternally. Otherwise, the not conclusive statement can be removed.

Minor comments: 

Figure legends for Fig.4h and Fig.8a: (lanes 1023-1025 and 1093-1096). Figure legend 4h (lanes 1023-1025) states the rescue of “the early developmental defect of MZnanog” Figure legend 8a (lanes 1093-1096) states the rescue of : “the dorsalization phenotype of MZnanog”. Menawhile, both figures show similar phenotypes, which only with with a lot of good will can be interpreted as a partial epiboly rescue in MZnanog by indicated constructs. Very confusing!

Lane 172 on – “knockdown of nanog leads to dorsalization and posteriorisation”. Here, the authors inject the previously published Nanog morpholino, used in two studies (39 and 52). They find, that the moderate dose of the morpholino (1.4 ng/embryo) results in dorzalisation of the embryos, while the head is normal: the low dose (0.5 ng/embryo) results in forebrain defects, which are similar to wnt8 overexpression. Strange enough, Perez-Camps et al (ref.52) who injected the very same morpholino in slightly different concentrations (1.6 ng/embryo and 0.4 ng/embryo), saw the forebrain defects only in the higher dose, in parallel with other multiple defects. This issue has to be discussed.

In Materials and Methods (595-602), the tcf4 mRNA used in the manuscript is missing, mxtx2, which was not used, is present. 

Lane 275 on: References to zebrafish Tcf7 and Tcf4 is not provided; it is unclear which genes were actually used. Zebrafish tcf4 is first expressed in organogenesis, tcf7 is expressed maternally but does not have known early function; it is unclear why these tcfs were selected. 

References:

Bellipanni, G., M. Varga, S. Maegawa, Y. Imai, C. Kelly, A. P. Myers, F. Chu, W. S. Talbot, and E. S. Weinberg. 2006. Essential and opposing roles of zebrafish beta-catenins in the formation of dorsal axial structures and neurectoderm. Development 133 (7):1299-1309.

Kawahara, A., and I. B. Dawid. 2000. Expression of the Kruppel-like zinc finger gene biklf during zebrafish development. Mech Dev. 97 (1-2):173-176.

Melby, A. E., D. Kimelman, and C. B. Kimmel. 1997. Spatial regulation of floating head expression in the developing notochord. Developmental dynamics : an official publication of the American Association of Anatomists 209 (2):156-165.

Ramel, M. C., G. R. Buckles, K. D. Baker, and A. C. Lekven. 2005. WNT8 and BMP2B co-regulate non-axial mesoderm patterning during zebrafish gastrulation. Developmental biology 287 (2):237-248.

Ramel, M. C., and A. C. Lekven. 2004. Repression of the vertebrate organizer by Wnt8 is mediated by Vent and Vox. Development 131 (16):3991-4000.

Varga, M., S. Maegawa, G. Bellipanni, and E. S. Weinberg. 2007. Chordin expression, mediated by Nodal and FGF signaling, is restricted by redundant function of two beta-catenins in the zebrafish embryo. Mech Dev. 124 (9-10):775-791. Epub 2007 Jun 2012.

Veien, E. S., M. J. Grierson, R. S. Saund, and R. I. Dorsky. 2005. Expression pattern of zebrafish tcf7 suggests unexplored domains of Wnt/beta-catenin activity. Dev Dyn 233 (1):233-239.

1)

Reviewer #3 (Bruno Reversade, signed review): 

Mudan He et al., provide a well-documented analysis of the role for Nanog in DV patterning in zebrafish using novel and genetic knockout animals. Their rescue experiments using variant Nanog constructs allowed to discriminate between the various functions of Nanog vis a vis MBT or DV axis patterning . The direct interaction and functional inhibition of maternal NANOG , via its N terminal domain towards TCF provides a compelling mechanism for the inhibition of beta-catenin-mediated WNT signaling on the ventral side. 

Overall I found this manuscript to be well written, coherently assembled, scientifically sound, and with genuine mechanistic intent. As such this is an significant study which merits publication. 

There are no major concerns since the author’s claims are supported by experiments which are performed with significant numbers and backed up by various orthogonal assays. 

Major comments:

1:Based on the authors conclusion, what would be the effect of removing endogenous maternal beta-Catenin via MO depletion in MZ null null Nanog. To what extent will this rescue the phenotype. This is a more physiological assay than overexpression of gsk3b or ck1a.

2:Since nanog is a potent transcription activator, it would be expected to have an RNAseq dataset of the transcriptome of MZ null Nanog embryos compared to wt before and after MBT to provide an unbiased profile of the differentially regulated genes. This would permit a more unbiased analysis to be done by the authors and other scientists. This should be deposited in the public domain.

3:The authors confidently show a dorsalization of MZ null Nanog embryos but do not look at the expression of the very genes/proteins that drive ventral fates i.e., BMPs whose levels also control the expression of dorsal genes (e.g. Chrd). This is lacking. The endogenous levels of expressions of Bmp2/7 and Admp by QPCR and WISH should be documented. In fact, one ligand that would be key to assess is the maternal TGFbeta Radar. Such data should be included in the revised manuscript to better describe the observed DV defects.

4:I would like to bring to the authors’ attention that by all means zebrafish and Xenopus have very similar ways of initiating early embryogenesis both in regards to MBT, germline patterning, etc. However, there does not seem to be a Nanog homologue in Amphibians. How can they reconcile or at least discuss this conundrum? 

Minor comments:

1:Correct syntax: Line 321: “Unlike that the ratio of active β322 catenin/total β-catenin was remarkably increased.”

2: Spelling mistake : 387 “Nanog is capable of binding with Tcf7, we generated serval deletion types of”

---

## [Decision Letter · Decision Letter 2]

16 Jun 2020

Dear Dr Sun,

Thank you for submitting your revised Research Article entitled "Nanog safeguards early embryogenesis against global activation of maternal β-catenin activity by interfering with TCF factors" for publication in PLOS Biology. I have now obtained advice from the original reviewers and have discussed their comments with the Academic Editor. 

Based on the reviews, we will probably accept this manuscript for publication, assuming that you will modify the manuscript to address the remaining points raised by the reviewers. Please also make sure to address the data and other policy-related requests noted at the end of this email.

We expect to receive your revised manuscript within two weeks. Your revisions should address the specific points made by each reviewer. In addition to the remaining revisions and before we will be able to formally accept your manuscript and consider it "in press", we also need to ensure that your article conforms to our guidelines. A member of our team will be in touch shortly with a set of requests. As we can't proceed until these requirements are met, your swift response will help prevent delays to publication.

*Copyediting*

*Published Peer Review History*

*Early Version*

*Submitting Your Revision*

Sincerely,

Di Jiang, PhD

PLOS Biology

ETHICS STATEMENT:

-- Please create a separate subsection entitled "Ethics Statement" and place it in the beginning of the Methods section. Please include all relevant information described below especially an approval number.

-- Please include the full name of the IACUC/ethics committee that reviewed and approved the animal care and use protocol/permit/project license. Please also include an approval number.

-- Please include the specific national or international regulations/guidelines to which your animal care and use protocol adhered. Please note that institutional or accreditation organization guidelines (such as AAALAC) do not meet this requirement.

-- Please include information about the form of consent (written/oral) given for research involving human participants. All research involving human participants must have been approved by the authors' Institutional Review Board (IRB) or an equivalent committee, and all clinical investigation must have been conducted according to the principles expressed in the Declaration of Helsinki.

DATA POLICY:

Regardless of the method selected, please ensure that you provide the individual numerical values that underlie the summary data displayed in the following figure panels as they are essential for readers to assess your analysis and to reproduce it: Figures 1DFG, 2DFGLIJN, 3DEG-K, 4DG-I, 5BCE, 6BCFG, 7F, 8C-E, S4B, S6AB, S7CEF, S9D. NOTE: the numerical data provided should include all replicates AND the way in which the plotted mean and errors were derived (it should not present only the mean/average values).

Reviewer remarks:

Reviewer #1: The authors have adequately addressed this reviewer's concerns by providing additional experimental data and clarifying the raised issues. The revised manuscript has be much improved. This review would recommend acceptance for publication in PLOS Biology. 

Reviewer #2 (Daria Onichtchouk, signed): I am impressed by the amount and quality of experimental work provided by authors in response to the requests on the first round of the review! All my concerns are fully addressed. The manuscript describes the previously unknown competition mechanism of Nanog and beta-catenin for TCF binding. The conclusions are well supported by clear experimental results, the logical flow is clear, the manuscript deserves publication. However, before publishing the manuscript, it is necessary to correct the multiple typos and language mistakes. Some (not all) of those are listed below:

Lane 68: presence (not present)

88: in (not through)

109: organizer (not organizers)

142: The nanog-deficient ES cells lost...

370: interfering with

457, 462: Word missing: These (experiments?)

466: does (not dose)

469: "To start with, we firstly identified" - reformulate this phrase

495: remove "that"

513: may interfere with

606-607: "...and there might be some negative-regulators on the coding sequence of C-terminal of Nanog" - the conclusion is unclear. 

Reviewer #3 (Bruno REVERSADE, signed): I am satisfied with the revisions.

The authors have taken at heart to respond to our concerns by direct experimentations.

---

## [Editor Report · Decision Letter 3]

3 Jul 2020

Dear Dr Sun,

On behalf of my colleagues and the Academic Editor, Mary C Mullins, I am pleased to inform you that we will be delighted to publish your Research Article in PLOS Biology. 

Early Version

PRESS 

Kind regards,

Alice Musson

Publishing Editor, 

PLOS Biology

on behalf of

Di Jiang, PhD,

Senior Editor

PLOS Biology